# Origins of lithium inventory reversibility with an alloying functional layer in anode-free lithium metal batteries

Lennart Wichmann [1], Shi-Kai Jiang[2], Johannes Helmut Thienenkamp [1], Marvin Mohrhardt[1], Bing Joe Hwang [2,3], Martin Winter [1,4] & Gunther Brunklaus [1] ✉

Alloying coatings are widely accepted to boost the reversibility of lithium inventory in anode-free cell configurations. While diminished capacity losses are evident from electrochemical data, their impact beyond decreasing the nucleation overpotential remains elusive. Herein, in situ [7]Li NMR spectroscopy is applied to differentiate capacity losses in pouch cells with representative electrochemical behavior. Next to an accelerated interphase formation, the alloying layer diminishes the formation of dead lithium deposits notably. In contrast to previous reports, the capacity lost to electronically insulated lithium deposits is not related to their tortuosity and surface area. Though alloy formation reduces the nucleation overpotential with coated copper, deconvolution of [7]Li NMR spectra as well as scanning electron microscopy evidence predominantly compact lithium deposits in the initial cycles and a similar increase of high-surface area morphologies with bare and coated copper negative electrodes. Instead of improved lithium deposit morphology, the enhanced reversibility with the alloying layer is bestowed by improved interfacial transport towards the end of lithium dissolution. These insights add to the mechanistic understanding of dead lithium formation, exploiting impedance spectroscopy in the discharged state as a valuable tool to assess the ability to dissolve lithium metal from a given substrate.

Novel cell concepts for electrochemical energy storage are required, since the energy densities for batteries relying on ion intercalation into carbonaceous negative electrodes are approaching their theoretical limits[1-6]. Reversible metal deposition instead of ion intercalation at the negative electrode allows boosting the achievable energy density on cell level. In conventional metal batteries invoking this concept, the ionic reservoir from the positive electrode is deposited on a pre-existing metal electrode upon charge[6,7]. Since the positive electrode can at most take up as much capacity upon discharge as previously

deposited upon charge, the metal layer represents an excess capacity reservoir occupying extra weight and volume, which already diminishes the achievable energy density of metal batteries[6]. To surpass the practical energy densities of ion intercalating negative electrodes, the excess metal electrode needs to be as thin as possible while still being compatible with large-scale battery production[8,9]. With the handling of thin metal films being challenging in terms of safety and mechanical stability, the so-called anode-free cell concept, solely exploiting the capacity reservoir available from the positive electrode, has received

[1]Helmholtz-Institute Münster, IMD-4, Forschungszentrum Jülich GmbH, Münster, Germany. [2]Nano-electrochemistry Laboratory, Department of Chemical Engineering, National Taiwan University of Science and Technology, Taipei, Taiwan. [3]Sustainable Electrochemical Energy Development (SEED) Center, National Taiwan University of Science and Technology, Taipei, Taiwan. [4]University of Münster, MEET Battery Research Center, Institute of Physical Chemistry, Münster, Germany. ✉e-mail: g.brunklaus@fz-juelich.de

increasing attention[10–12]. Here, the volume and mass of the negative electrode are minimized by using the negative electrode current collector as a substrate to form the metal electrode in situ, achieving the maximal energy density possible with the respective positive electrode. Furthermore, cell assembly is simplified by abandoning the metal electrode while also limiting potential safety hazards and costs[11,13]. These benefits, however, come at the expense of the battery's lifetime since the abundant excess capacity of the metal reservoir at the negative electrode no longer replenishes capacity losses occurring upon cell operation. Thus, irreversible phenomena and losses of capacity are expected to immediately diminish the actual energy density as well as cycle life[14]. Currently, the limited lifetime of anode-free cells due to fast capacity decay restricts the application of this intriguing cell concept.

Two distinct processes have previously been identified as major contributors to the capacity losses occurring in anode-free lithium metal batteries (anode-free LMBs). In particular, lithium is found to be irreversibly consumed by the (re-) formation of the interphases between the electrolyte and the positive (CEI) and negative (SEI) electrode, yielding organic and inorganic lithium salts[15,16]. Since non-passivated surfaces are created continuously throughout cycling, especially at the negative electrode, losses of lithium inventory due to CEI and SEI formation are known to steadily consume available cell capacity. In addition to electrode passivation, electronically insulated metallic deposits at the negative electrode that are no longer participating in redox reactions (often referred to as inactive or dead lithium metal) are known to also contribute to capacity fading of anode-free LMBs[15,17–19]. Strategies such as tailoring the electrolyte, surface patterning, or functionalizing the negative electrode by artificial coatings have been employed to mitigate capacity losses due to irreversible processes in anode-free LMBs[20–23]. Upon increasing the salt concentration within the electrolyte, eventually yielding (localized) high-concentrated electrolytes (LHCEs), formation of predominantly inorganic solid electrolyte interface (SEI) layers has been reported as a feasible approach to improve long-term stability[24]. In the case of current collector functionalization, elements alloying with lithium (i.e. forming intermetallic phases) are widely applied since they lower the overpotential for initial nucleation onto copper, which is thought to enable a more homogeneous lithium deposition[25–27]. While the improved reversibility of lithium inventory due to these approaches is readily displayed by comparing the respective electrochemical cell performances, its origins remain unclear. In order to uncover details of the mechanisms behind increased cell longevity and enable knowledge-driven improvement of materials for anode-free cell systems, the observable capacity losses need to be differentiated, revealing which irreversible processes were diminished or limited.

Due to its element specific and non-destructive properties, [7]Li nuclear magnetic resonance (NMR) spectroscopy represents a precise analytical method to quantify the formation of dead lithium deposits in anode-free lithium metal batteries[15,18,19,28,29]. Here, ionic and metallic lithium species that are present within cells are well separated in terms of their respective chemical shifts due to the additional electron, which allows for a straightforward detection of lithium metal[30–32]. Knowing the amount of capacity lost to dead lithium fractions, the remaining capacity loss displayed in electrochemical data can be attributed to SEI and CEI (re)formation[15,18]. Furthermore, [7]Li NMR spectroscopy contains information on lithium metal deposit morphology since the shape, density and orientation of lithium deposits to the magnetic field is reflected by their [7]Li chemical shifts[30–33]. Thus, not only the amounts of dead lithium metal but also fractions of dense, mossy, and dendritic lithium deposits can be identified. Unlike conventional methods utilized to quantify dead lithium or asses the morphology of lithium metal deposits (i.e. scanning electron microscopy or titration gas chromatography), the results obtained via NMR spectroscopy are not tainted due to cell disassembly, unrepresentative spots or side reactions[19,34].

However, for insights obtained via [7]Li NMR spectroscopy to be representative of application-oriented anode-free lithium metal batteries, cells compatible with the individual geometry of the NMR probe need to fulfill two criteria. On the one hand, the electrochemical behavior of such cells needs to be comparable to commercially available cell formats (e.g. coin or pouch cells) while on the other hand, the cell chemistry (i.e. combination of positive and negative electrode) should be of relevance for applications with respect to the projected energy densities. Previous efforts that distinguish origins of capacity losses in anode-free lithium metal batteries via [7]Li NMR either utilized Cu∥Li or Cu∥lithium iron phosphate (LFP) cells. Furthermore, only initial cycles were considered, and the electrochemical behavior of the NMR compatible cell format was not validated against conventional cell formats. Due to the lower specific capacity and operating voltage of LFP compared to Nickel-Manganese-Cobalt-Oxides (NMC) positive electrodes, LFP-based anode-free lithium metal batteries achieve lower energy density than state-of-the-art NMC-based lithium-ion batteries while having a lower lifetime, which renders this cell chemistry unfavorable for higher energy density applications[10,35]. Electrochemically, the insights obtained with LFP positive electrodes have limited transferability to NMC-based anode-free cell systems. The higher upper cut-off voltage alters the amount of CEI formation as well as its composition and may exceed the electrochemical stability window of electrolytes applied in Cu∥LFP cells[36–38]. Furthermore, the occurrence of transition metal dissolution is known to induce inhomogeneous lithium deposition or lithium dendrite formation and thus results in accelerated capacity fading during cycling[39–41]. Most importantly, NMC directly impacts the deposition and dissolution of lithium metal at the negative electrode since an increased activation energy for lithium diffusion towards the end of NMC re-lithiation limits lithium dissolution and creates an excess lithium metal reservoir in the initial cycle, which is not present in the case of LFP cathodes[42,43].

We herein utilize in situ [7]Li NMR spectroscopy on NMR compatible pouch cells displaying coin cell like electrochemical performance to deconvolute contributions to the capacity fading in anode-free lithium metal batteries operated with localized high-concentration electrolytes (LHCE). Hereby, the improved cell longevity bestowed from an alloying-type functional layer in NMC-based anode-free lithium metal batteries is unraveled to originate from an accelerated stabilization of the SEI as well as notably diminished formation of dead lithium deposits. While the global lithium deposit morphology and its evolution throughout cycling are similar for both bare and coated copper negative electrodes, enhanced interfacial transport due to irreversible alloy formation is determined as the dominant reason for reduced dead lithium formation.

## Results

### Quantification of dead lithium fractions in NMC-based cells

For quantitative NMR studies, knowledge of the recycling delay and optimal pulse length are typically required (Supplementary Fig. 1a, b). When employing metallic electrodes, impairment of signal penetration throughout the whole electrode due to the skin-depth effect may also limit the quantitative nature of NMR spectroscopy. With the herein utilized magnetic field strength of 4.7 T, the penetration depth of radiofrequency pulses invoked to excite nuclei of interest is calculated to be 14.7 μm[32]. Thus, quantitative assessment of lithium metal batteries, even with thin lithium metal electrodes, can be challenging[32]. Ex situ as well as operando experiments on anode-free Cu∥NMC622 cells render lithium deposits of up to 2.5 mAh cm$^{-2}$ quantitatively accessible, since the integral of the deposited metallic lithium increases linearly with charge capacity (Supplementary Fig. 1c, d). Note that the signal attenuation is only negligibly impacted when comparing the negative electrodes comprised of purely lithium metal or lithium metal on a copper current collector (Supplementary Fig. 2a).

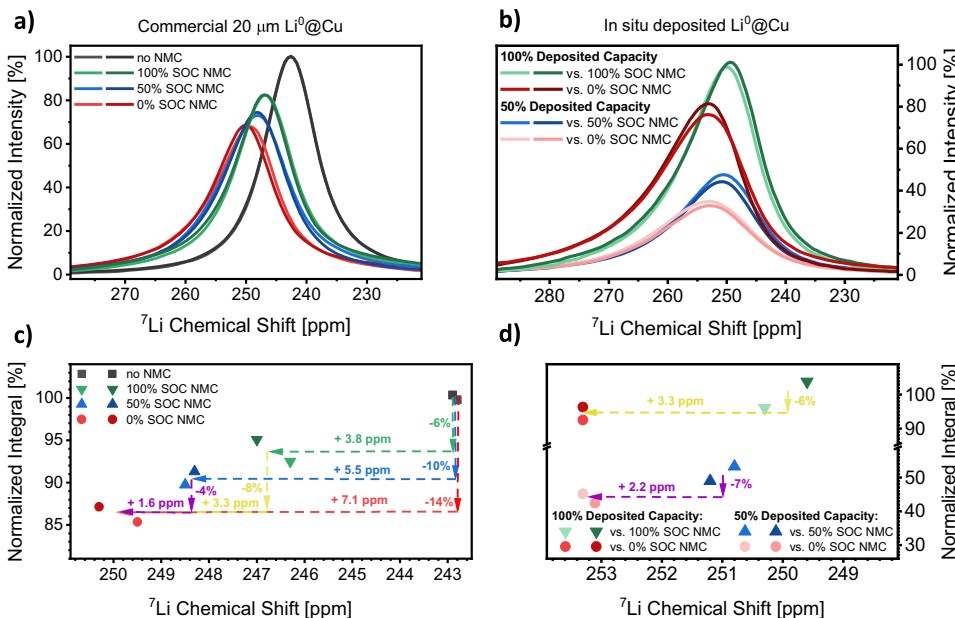

**Fig. 1 | Calibration of NMC622 impact on $^7$Li NMR spectra using NMR compatible pouch cells. a** $^7$Li NMR spectra of 20 μm lithium metal on 10 μm copper electrodes in vicinity of NMC622 positive electrodes with varying state of charge (SOC). The intensities are normalized to lithium metal on copper electrodes without a NMC positive electrode. **b** $^7$Li NMR spectra of lithium metal deposited on negative copper electrodes in vicinity of NMC622 positive electrodes with varying SOC. The intensities are normalized to 100% deposited capacity with a NMC622 positive electrode at 100% SOC. In (**a**) and (**b**), slightly darker colors indicate the reproduction of results by repeating measurements once with a new, nominally identical cell. **c** Evaluation of (**a**) after integration, normalized to the integral of lithium metal electrodes without a NMC positive electrode. **d** Evaluation of (**b**) after integration, normalized to the integral of 100% deposited capacity with a NMC622 positive electrode at 100% SOC. In (**c**) and (**d**), the changes in $^7$Li integral and chemical shift of commercial and deposited lithium metal when the NMC622 SOC is altered from either 100% to 0% or 50% to 0% are marked in yellow and purple, respectively. Electrolyte: LHCE.

To accurately quantify the amount of dead lithium via $^7$Li NMR spectroscopy in NMC-based anode-free lithium metal batteries, the impact of the positive electrode on the spectroscopic as well as electrochemical observables has to be understood. Comparing $^7$Li NMR spectra for identical pristine lithium metal electrodes (20 μm Li$^0$ on 10 μm copper) in both the absence and presence of NMC622 electrodes already indicates differences in the $^7$Li NMR chemical shifts as well as integrals and widths of the peaks attributed to lithium metal species (Fig. 1a, c). This is due to the paramagnetism of transition metals within NMC, which increases the local magnetic fields in the vicinity of the lithium metal deposits, thereby causing a shift towards higher frequencies[44]. Notably, this effect depends on the positive electrodes state-of-charge (SOC), since the magnetic properties of the transition metals are determined by their corresponding oxidation states[45]. NMC622, in pristine condition, exerts the strongest increase of the chemical shift (+7.1 ppm) in the case of the dense, as-manufactured lithium electrodes. Upon charge of the battery, which corresponds to NMC delithiation and a reduced paramagnetism due to Ni$^{2+}$ being oxidized to Ni$^{3+}$ and Ni$^{4+}$, the impact from the positive electrode on the chemical shift of lithium metal decreases.

The reduced lithium metal $^7$Li NMR integral due to the NMC positive electrode also correlates with its SOC. While both, pristine and charged NMC622 electrodes, diminish the lithium metal integrals, a more pronounced decrease is evident for uncharged (0% SOC) NMC electrodes. This can be attributed to the non-ideal excitation of the NMR sample since the local paramagnetism induces inhomogeneities in both the static ($\mathbf{B_0}$) as well as applied radiofrequency magnetic field ($\mathbf{B_1}$) invoked to excite the samples. These effects also occur for in situ deposited lithium metal (Fig. 1b, d), which will be analyzed by $^7$Li NMR spectroscopy in later sections. Contrary to commercial lithium metal electrodes of 20 μm thickness, the detection of in situ deposited lithium metal up to capacities of 2.5 mAh cm$^{-2}$ via $^7$Li NMR spectroscopy is not impaired by the skin depth effect, which describes finite penetration of electromagnetic fields in electronic conductive materials[46]. With in situ deposited lithium metal, the peak integral deterioration due to paramagnetic non-idealities is similar compared to commercial lithium electrodes (6% vs. 8% and 7% vs. 4% intensity decay when substituting 100% and 50% SOC NMC with pristine NMC at 0% SOC, respectively). Also, higher chemical shifts due to the SOC-dependent paramagnetism of NMC622 electrodes are of similar magnitude for in situ deposited lithium metal compared to commercial lithium electrodes (+3.3 ppm when substituting 100% SOC NMC with 0% SOC NMC in both cases and +2.2 vs. +1.6 ppm substituting 50% SOC NMC with 0% SOC NMC, respectively).

While the calibration with in situ deposited lithium benefits from quantitative detection of lithium metal, potential material losses or damage to the lithium morphology upon cell disassembly might impact the results. Though $^7$Li NMR measurements of separators obtained from cell disassembly did not show residual lithium metal remaining within the separator (Supplementary Fig. 2b), lithium metal might still be scratched off by contact with tweezers or scalpel used to disassemble the cells. Nevertheless, both sets of experiments exhibit sufficient agreement and reveal that the respective integrals for lithium metal species have to be corrected for the impact of the positive electrodes depending on their state-of-charge. Note that a separate calibration (Supplementary Fig. 3) is required for each individual cell setup since the demonstrated effects are expected to depend on the inter-electrode distance, positive electrode mass loading and stoichiometry.

In addition to the spectroscopic implications for $^7$Li NMR analysis, NMC positive electrodes also alter the electrochemical procedure for quantification of dead lithium fractions. Even with a seemingly limitless lithium metal reservoir available in conventional lithium metal batteries, substantial apparent irreversible capacity is observed in the initial cycle with NMC positive electrodes[42]. Accordingly, the initial Coulombic efficiency when operating NMC-based LMBs is lower

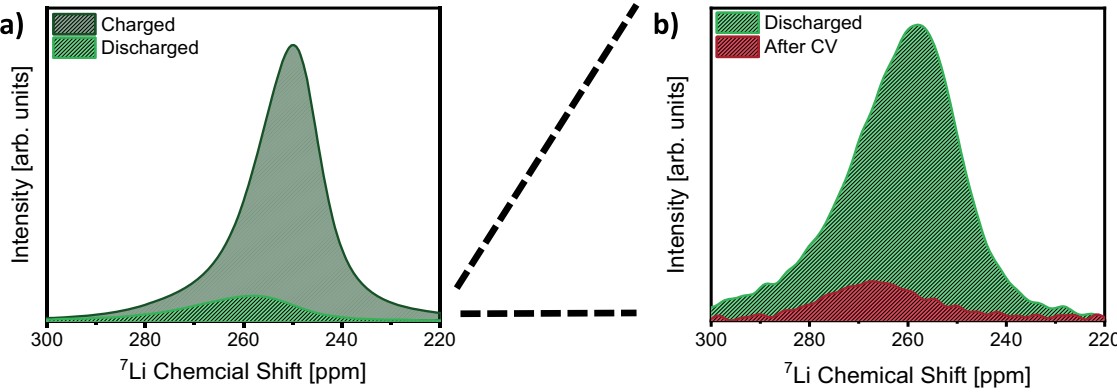

**Fig. 2 | Exemplary $^{7}$Li NMR spectra for the quantification of dead Li$^{0}$ in Cu‖NMC cells. a** Comparison of metallic $^{7}$Li NMR signals for NMC-based anode-free lithium metal batteries after charge and discharge. **b** Differentiation of metallic lithium remaining after constant current and constant voltage (CV) discharge into excess and dead lithium species via $^{7}$Li NMR. Electrolyte: LHCE.

compared to other positive electrode materials such as LMNO or LFP (Supplementary Fig. 4a). Instead of increased irreversible processes with NMC positive electrodes, lower recovery of charge capacity in the case of NMC electrodes is due to a kinetic hindrance upon approaching complete NMC re-lithiation[47–50]. Thus, more positive electrode de-lithiation than re-lithiation typically occurs in the initial cycle of NMC-based (lithium metal) batteries. As a consequence, excessive lithium (metal) remains at the negative electrode, which may be consumed at later stages of cell cycling. To quantify the amount of dead lithium fractions in NMC-based anode-free LMBs, the excess lithium metal at the negative electrode needs to be removed, leaving behind dead lithium as the only metallic lithium species. Since the limitation of lithium stripping with NMC is of kinetic nature, re-lithiation of NMC can be achieved by imposing a slow discharge step at a constant voltage (CV discharge)[42]. This enables the complete dissolution of all remaining reversible lithium metal deposits and renders irreversible phenomena instead of kinetically-driven NMC re-lithiation issues the bottleneck for lithium metal dissolution, uncovering the reversibility of lithium inventory. Consequently, lithium metal in the discharged state can be differentiated into active (but positive electrode limited) excess lithium and dead lithium, which remains after the CV step (Fig. 2). In turn, the $^{7}$Li NMR signal for metallic lithium after the CV discharge step can be utilized to quantify the amount of dead lithium formed in Cu‖NMC622 cells after a certain number of cycles. Comparing the $^{7}$Li NMR signal in the discharged state after the 1$^{st}$ cycle in the case of NMC-, LFP- and LMNO-based anode-free LMBs (Supplementary Fig. 4b) indeed emphasizes the necessity of performing a constant voltage discharge step when using NMC positive electrodes. Here, the signal for dead lithium metal observed via $^{7}$Li NMR, which should be mostly determined by the negative electrode and thus independent of the positive electrode, is only of comparable intensity for all considered positive electrodes when carrying out a CV discharge step with cells containing NMC positive electrodes.

A comparison of the Coulombic efficiencies of anode-free and conventional lithium metal batteries with the different positive electrodes after a constant voltage discharge step (Supplementary Fig. 5) reveals that solely NMC positive electrodes display a significant kinetically limited capacity in anode-free configurations. While NMC-based lithium metal batteries exhibit almost identical initial Coulombic efficiencies independent of lithium metal or copper being the negative electrode, these differ for LFP and LNMO depending on the negative electrode. This exemplifies that solely for NMC-based cells, the discharge process can be limited by the positive electrode. Note that LFP displays minor kinetically limited capacity with lithium metal as negative electrode (2.4% of the initial charge capacity). However, this reservoir is consumed almost completely in the initial cycle of an anode-free cell setup, as perceptible by a decrease in Coulombic efficiency (reversible capacity) compared to a conventional lithium metal battery (Supplementary Fig. 5).

## Differentiation of capacity losses

While operation of anode-free lithium metal batteries with localized high-concentration electrolytes has been demonstrated to yield anion-derived interfaces with superior passivation and Coulombic efficiencies exceeding 99%[24], the longevity of these Cu‖NMC622 cells still remains rather limited (80% state-of-health after 35 cycles, Fig. 3a). We herein utilize the as-developed $^{7}$Li NMR protocol to understand the current limitations of this benchmark system. Furthermore, the positive impact on the reversibility of lithium inventory stemming from an alloying coating on copper is also explored. Here, a previously introduced[51,52] artificial coating comprised of SrF$_2$ nanoparticles, PVDF-HFP and a metallic tin-layer (referred to as coated Cu) is selected as a state-of-the-art electrode modification of negative electrodes, considering that the combination of lithium-alloying metals and polymeric compounds afforded improved Coulombic efficiencies in anode-free lithium metal batteries in multiple cases[27,51–53]. Upon initial charge of the cells, Sn and Sr species may form alloys with lithium, while the fluoride species allow for the formation of LiF-rich and durable SEI layers[51,52]. The polymeric PVDF-HFP was demonstrated to inhibit dissolution of SEI species into the electrolyte, thereby diminishing continuous interphase formation[53]. With our operating conditions, replacement of bare copper by coated copper improves the cycle life by >40% in coin cells (80% state-of-health after 50 instead of 35 cycles, Fig. 3a). To substantiate previous claims by differentiating capacity losses via $^{7}$Li NMR spectroscopy, a cell setup suitable for $^{7}$Li NMR measurements is required. Here, conventional cell housings (e.g. coin cell cases or pouch foil) have to be replaced due to interferences with the measurement technique (that is, limited penetration of the radio frequency (RF) pulses and signal damping) while maintaining a comparable cycling behavior to derive data that represents application-oriented cells. A comparison of the electrochemical behavior of coin cells and custom-made pouch cells (Supplementary Fig. 6) suitable for in situ NMR analysis underlines the practicability of the developed cell concept. Though specific discharge capacities are lower in the case of the NMR pouch cells, the Coulombic efficiencies and evolution of state-of-health (SOH) upon cycling display a high agreement between commercial and custom cell housings (Fig. 3a).

With the $^{7}$Li NMR integral for lithium metal dissolved upon discharge (i.e. difference between the lithium metal integral in charged and discharged state, Fig. 2) being representative of the electrochemical discharge capacity, conversion values to translate $^{7}$Li NMR integrals into capacities can be deduced, allowing to quantify the dead

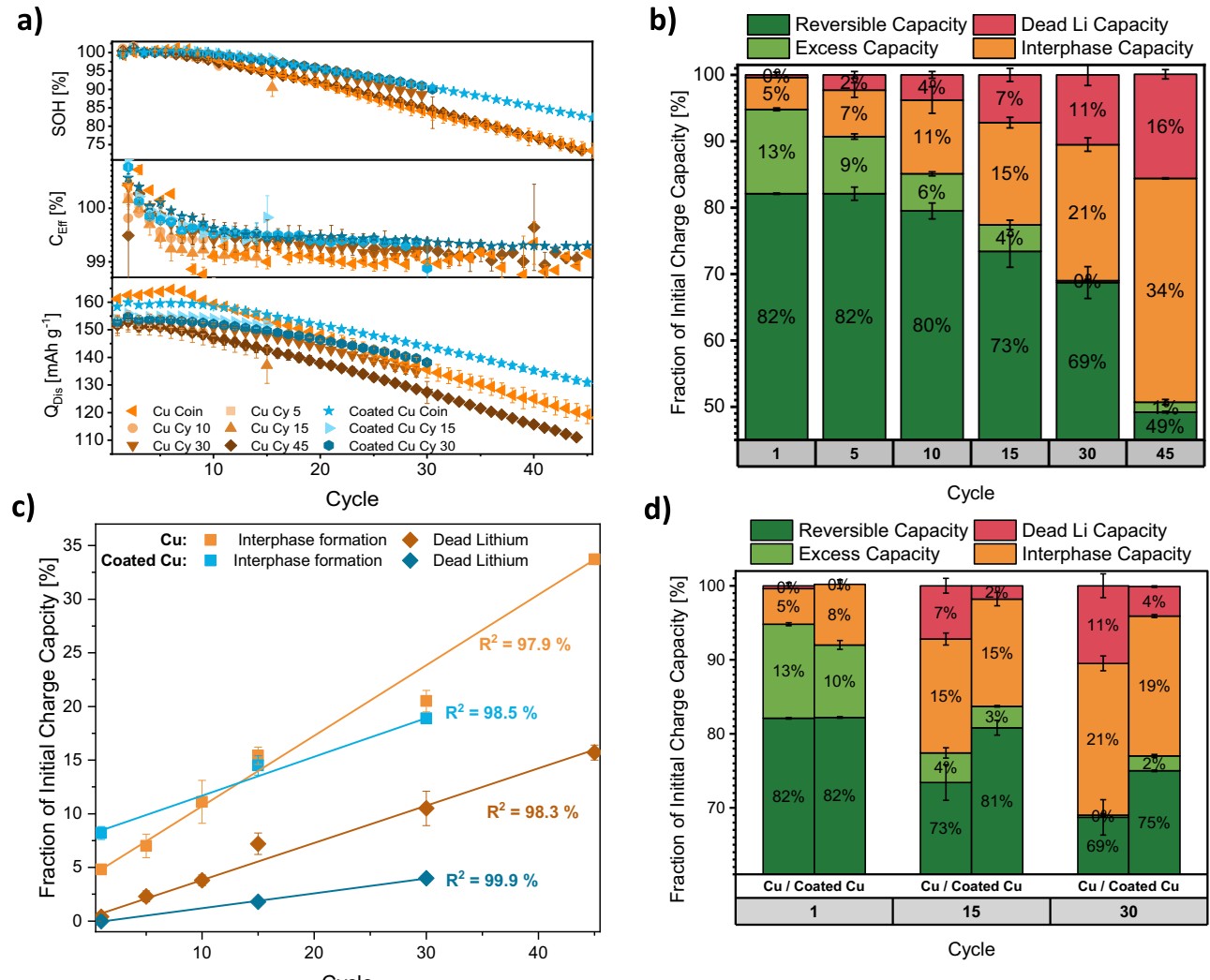

**Fig. 3 | $^7$Li NMR-based quantification of capacity losses for Cu||NMC622 and coated Cu||NMC622 NMR pouch cells. a** Evolution of discharge capacity ($Q_{Dis}$), Coulombic efficiency ($C_{Eff}$), and state-of-health (SOH) for Cu||NMC622 and coated Cu||NMC622 coin and NMR pouch cells. **b** Cumulative allocation of the initial charge capacity to reversible capacity, excess capacity, interphase capacity, and dead lithium capacity for Cu||NMC622 NMR pouch cells. **c** Linear fit of the capacity

losses for both negative electrodes with increasing cycle number. **d** Cumulative allocation of the initial charge capacity to reversible capacity, excess lithium, interphase capacity, and dead lithium for Cu||NMC622 and coated Cu||NMC622 pouch cells. In all graphs, error bars represent standard deviation between at least two nominally identical measurements with different cells. Electrolyte: LHCE.

lithium capacity from the $^7$Li NMR integral detected after the CV discharge step (Supplementary Fig. 7). Assuming that the remaining capacity losses are due to interphase (re)formation, the initial charge capacity can be allocated into reversible capacity, kinetically limited (excess) capacity, dead lithium and capacity cumulatively lost to the (re)formation of the interphases (Fig. 3b)[15,18]. For the reference system using non-functionalized copper, (re)formation of interphases constitutes the dominant cause for the observable capacity decay. The formation of dead lithium fractions makes up a lower fraction of the observed capacity losses, agreeing with previous reports that utilize titration gas chromatography to quantify dead lithium fractions in anode-free systems using LHCEs[54–56]. Nevertheless, dead lithium species accumulate over time and have a notable contribution to the loss of lithium inventory. Both, losses of cell capacity due to the formation of interphases and dead lithium fractions, accumulate linearly with the number of cycles (Fig. 3c). When utilizing coated copper, the formation of SEI and CEI layers still is the main contributor to the observed capacity decay upon cycling (Fig. 3d). Compared to bare copper, the loss of capacity due to interphase formation is higher initially but levels of afterwards. Thus, slightly less capacity has been lost to interphase

formation in cells utilizing coated copper negative electrodes after 30 cycles ($18.9 \pm 0.2\%$ instead of $20.5 \pm 1.0\%$). We interpret this behavior as an accelerated interphase formation and stabilization due to the functional coating. Though differentiation of capacity losses stemming from interphase formation at the positive or negative electrode is not feasible from the method presented herein, we suggest that the improved interphase formation in the case of coated copper is associated with the interphase at the negative electrode. This is because the electrolyte, cycling procedure, and positive electrode (which are considered the decisive factors for the interface formation at positive electrodes) are identical in both cell chemistries[28,36]. Furthermore, a lower rate of SEI dissolution and subsequent Li$^0$ corrosion was previously reported by applying a composite coating comprised of PVDF as a low-solubility polymer and embedded alloying metal fluorides, which is in agreement with the interfacial capacity losses quantified herein via $^7$Li NMR spectroscopy[53]. Besides the accelerated establishment of a stabilized SEI layer, allocation of initial charge capacity also reveals considerably less formation of dead lithium fractions with coated negative electrodes. Compared to bare copper, the capacity that is cumulatively lost to the formation of

dead lithium after 30 cycles is reduced by ≈60% in the presence of the functional coating.

In contrast to the loss of lithium due to interphase formation, a reduced amount of dead lithium formation when using the coated copper is already observable after 15 cycles. With dead lithium originating from the complete electronic insulation of lithium metal deposits by SEI layers[57], the extent of capacity losses due to both phenomena is interconnected. Thus, the decreased dead lithium formation with coated copper most likely also contributes to the extenuated loss of capacity due to SEI formation and vice versa. In order to guide future development of novel materials that boost the reversibility of lithium inventory in anode-free cell configurations, the mechanism that enables higher reversibility of lithium metal dissolution when operating cells with coated copper, particularly forming fewer dead lithium metal deposits, will be elucidated in the following sections.

## Correlation of capacity losses and lithium morphology

Since electronically insulated lithium deposits are thought to originate from a fracture of high surface area lithium deposits (HSAL, comprising mossy and dendritic morphologies), the extent of dead lithium formation is expected to correlate with the structure and porosity of lithium metal deposition (i.e. its morphology)[58,59]. The same applies for the amounts of lithium lost due to SEI formation, since the surface area of lithium metal exposed to the electrolyte is considered a decisive factor[58,60]. Thus, a compact deposition of lithium metal with a low surface area is expected to diminish capacity losses to both phenomena. Due to bulk magnetic susceptibility effects arising from the porosity, shape and orientation of metallic lithium deposits, their chemical shift and peak width can be evaluated to assess the lithium deposition morphology. The relations between $^7$Li NMR chemical shifts for mossy and dendritic lithium metal morphologies are already established for Li∥Li and Cu∥Li cells[15,31–33]. To evaluate the lithium deposit morphology in full cells, paramagnetic contribution of NMC positive electrodes to the local magnetic field have to be considered (Fig. 1). Since the $^7$Li NMR chemical shifts of commercial, dense lithium electrodes on a copper current collector increases from 242.9 ± 0.1 ppm to 246.7 ± 0.3 ppm due to the presence of NMC622 at 100% SOC, the chemical shift range of 246.5 ± 2.5 ppm is assigned to dense or compact lithium metal deposits. For mossy and dendritic lithium structures, previously defined chemical shift boundaries[30,31,33] are also adjusted by ≈3.5 ppm, taking the positive electrode's paramagnetism into account. To assess the contributions of each morphology to the total lithium metal deposition, the respective $^7$Li NMR peaks are deconvoluted with three Voigt functions having restricted chemical shifts representative of dense, mossy, and dendritic lithium metal species. Furthermore, each individual peak width is constrained to not exceed the width observed for commercial lithium metal electrodes in the vicinity of NMC622 electrodes at 100% SOC in the calibration measurements (Fig. 1a).

For the initial deposition of lithium metal on bare and coated copper, $^7$Li NMR spectroscopy (Fig. 4a, b) and scanning electron microscopy (SEM, Fig. 4c, d) consistently indicate pre-dominantly dense lithium deposits. Nevertheless, high surface area (i.e. mossy and dendritic) lithium metal deposits are also observable on both negative electrodes. The good agreement of NMR and SEM data is also validated by comparing identical electrodes with both methods in the 1st and 30th cycle (Supplementary Fig. 8a). As readily perceptible from the higher chemical shifts displayed in $^7$Li NMR spectra (Fig. 4a), the proportion of mossy and dendritic structures increases throughout cycling for both cell chemistries. Towards the end of either cell's cycle life, these morphologies comprise the major fraction of deposited lithium metal (Fig. 4b). The observed deterioration of lithium deposit morphology (Supplementary Fig. 8b) throughout cycling could be ascribed to both, the accumulation of electrolyte decomposition products and dead

lithium fractions, increasing the inhomogeneity of the interphase at the negative electrode[61,62]. As a consequence, preferential spots for the deposition of lithium metal are expected, increasing the local current density and thus surface area of deposits[58,60].

Despite the quantified differences in dead lithium formation and stabilization of SEI layers between cells with coated and bare copper, the initial morphology of metallic lithium deposits as well as their evolution throughout cell cycling is highly similar. Based on the nucleation overpotential observed for bare copper, which is alleviated by the functional coating (Supplementary Fig. 9), notable differences in high-surface-area lithium deposits would be expected[51]. Decreasing the nucleation overpotential by means of alloying has been reported to reduce the amount of mossy and dendritic structures[27,51–53,63,64], since the quantity of lithium nuclei scales with the cubic power of the nucleation overpotential, while an inversely proportional relation has been determined for the size of nuclei[65]. When utilizing carbonate-based electrolytes (rather than LHCEs), this relation holds true and a reduced nucleation overpotential with coated copper (Supplementary Fig. 9) translates into lower $^7$Li NMR chemical shifts for the lithium metal deposition after initial charging (Supplementary Fig. 10a), indicating lithium deposition via less nuclei with larger size and thus reduced surface area of lithium deposits (Supplementary Fig. 10b). When employing the LHCE instead of carbonate-based electrolytes, the initial morphology of lithium deposits is drastically improved for both, bare and coated copper (comparing Fig. 4b and Supplementary Fig. 10b). Albeit bare copper also displaying a nucleation overpotential with this electrolyte, no differences for the morphology of deposited lithium metal after the initial charge are observed among the substrates, neither via top-view SEM nor $^7$Li NMR deconvolution. Cross-section SEM images also display highly dense lithium deposits on bare and coated copper (Supplementary Figs. 11 and 12, respectively). However, upon inspection of the bottom layer of lithium metal deposits remaining after the initial discharge process, differences in lithium metal morphology in case of bare and coated copper are also observed when employing LHCE (Supplementary Figs. 13 and 14). While the nucleation layer on bare copper is comprised of rather mossy structures, more compact lithium deposits are observed on coated copper, agreeing with theory on lithium nucleation and its overpotential[65]. Note that due to the kinetic limitation of NMC relithiation and the minor formation of dead lithium fractions quantified in the 1st cycle, the lithium deposits remaining after initial discharge are mostly reversible and can be considered regular rather than dead lithium deposits.

Our assessment of lithium deposit morphology using $^7$Li NMR and cross-section SEM emphasizes that the nucleation overpotential is decisive for the initial lithium nucleation on the substrate, but not necessarily for the subsequent deposition and growth of lithium on lithium. In the LHCE, predominantly dense lithium deposits grow irrespective of the nucleation layer, while differences in nucleation overpotential propagate throughout subsequent lithium growth when using a carbonate-based electrolyte. Comparing the variation of the electrolyte to the impact of alternating the substrate and its nucleation overpotential, a decrease in HSAL is achieved more effectively by tailoring the electrolyte rather than the substrate. Taking these observations into account, we suggest that the properties of the electrolyte and its SEI layers play a more dominant role for the resulting global lithium deposit morphology by impacting the subsequent growth of lithium metal on lithium metal. Since not only the $^7$Li NMR chemical shift for all lithium deposits after charge (Fig. 4) but also isolated metallic lithium deposits (Supplementary Fig. 7) on bare and coated copper are similar throughout cell cycling, whereas quantified amounts of dead lithium differ notably, we conclude that fracture of high-surface-area lithium deposits and global differences in lithium morphology are not the decisive factor for enhanced reversibility of lithium dissolution when utilizing coated copper. Nevertheless, a more

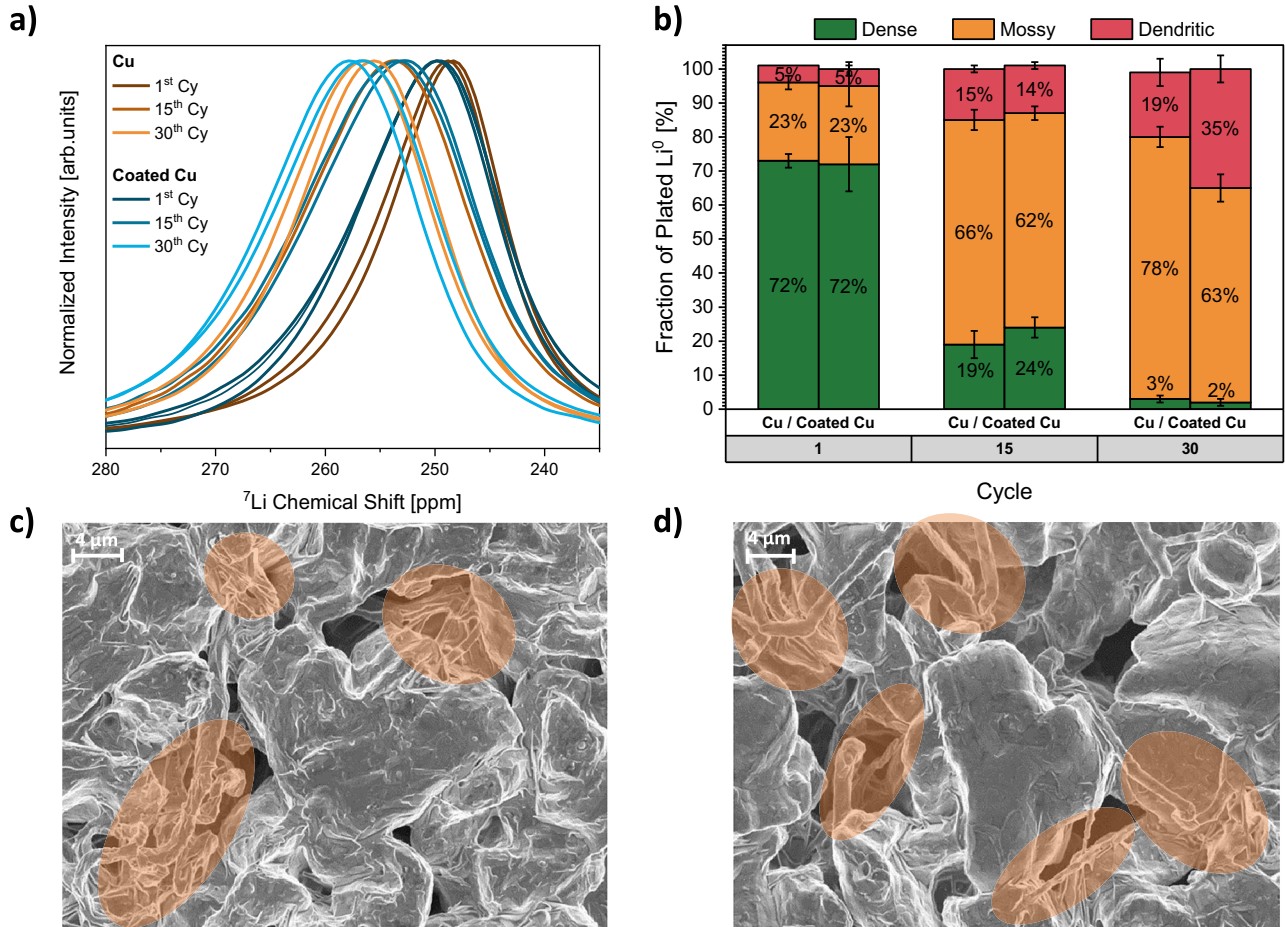

**Fig. 4 | Assessment of lithium deposit morphology in Cu||NMC622 and coated Cu||NMC622 cells. a** Metallic region of $^7$Li NMR spectra throughout cycling for Cu||NMC622 and coated Cu||NMC622 cells. Each measurement was reproduced at least once with a nominally identical, new cell. Both spectra are plotted. **b** Comparison of Li$^0$ morphology determined by deconvolution of NMR spectra throughout cycling for Cu||NMC622 and coated Cu||NMC622 cells. Error bars represent standard deviation between at least two nominally identical measurements repeated with different cells. **c** SEM image for the initial Li$^0$ deposition on bare copper in Cu||NMC622 coin cells charged to 100% SOC. Lithium deposits displaying high surface area are marked in orange. **d** SEM image for the initial Li$^0$ deposition on coated copper in coated Cu||NMC622 coin cells charged to 100% SOC. Lithium deposits displaying high surface area are marked in orange. Electrolyte: LHCE.

compact nucleation layer observed with coated copper (Supplementary Figs. 13 and 14) cannot be ruled out to also contribute to fewer electronically insulated lithium deposits.

### Mechanism of dead lithium formation in LHCEs

With the morphology of lithium metal deposits investigated, detailed characterization of interfacial processes may provide further mechanistic insights regarding the origin for improved lithium dissolution on coated copper. Electrochemical impedance spectroscopy (EIS) was performed on NMR pouch cells in parallel to NMR measurements. The validity of the obtained EIS data is demonstrated by showing linearity, stability and causality in the supporting information (Supplementary Figs. 15 and 16). Note that the limited frequency range of Kramers-Kronig compliant data introduces uncertainties for fully reliable fitting and especially quantitative extraction of resistances. Nevertheless, qualitative analysis of trends in impedance evolution based on equivalent circuit fits for visualization (Supplementary Fig. 17) is still applicable. Displaying impedance data for Cu||NMC622 and coated Cu||NMC622 cells in charged and discharged state as Nyquist plots (Fig. 5a), two semi-circles representing processes of different characteristic relaxation times can be observed. In agreement with previous EIS interpretations, high-frequency processes are assigned to grain boundary resistance and capacitance, which are either between the redox-active material (NMC or Li$^0$) and the current collector (Al or Cu) or neighboring (NMC and Li$^0$) particles for both

electrodes[66–69]. The semi-circle at lower frequencies reflects processes at electrode-|electrolyte-interfaces[66–69]. Compared to the charged state, cell impedance in the discharged state increases for both processes (Fig. 5a). Since the resistance is inversely proportional to the actual surface area of the electrode but the data can only be normalized to the macroscopic electrode area based on its dimensions, we attributed this trend to a decreasing surface area of lithium metal at the negative electrode upon electrochemical lithium dissolution[70]. Also, increased impedances at low SOCs are known for NMC-based LIBs[71,72], which rationalizes the observed SOC dependence for the positive electrode as well.

Throughout cycling, cell impedances in the charged state decrease in a similar manner with both negative electrodes (Fig. 5a and Supplementary Fig. 18a). With a deterioration of lithium deposit morphology throughout cycling reflected in $^7$Li NMR spectra (Fig. 4a, b), the decrease of cell resistance in the charged state upon cycling might be correlated with the increase in actual electrode surface area for both negative electrodes. In the discharged state, the evolution of cell impedance throughout cycling is vastly different with both negative electrodes (Fig. 5b and Supplementary Fig. 18b). Here, cells using bare copper negative electrodes exhibit a drastic increase of grain boundary as well as interfacial impedances after 15 cycles. Cells with coated copper display a more pronounced increase in grain boundary resistance but less resistive interfacial processes. While the respective semi-circle is not fully resolved towards the lower frequency limit of

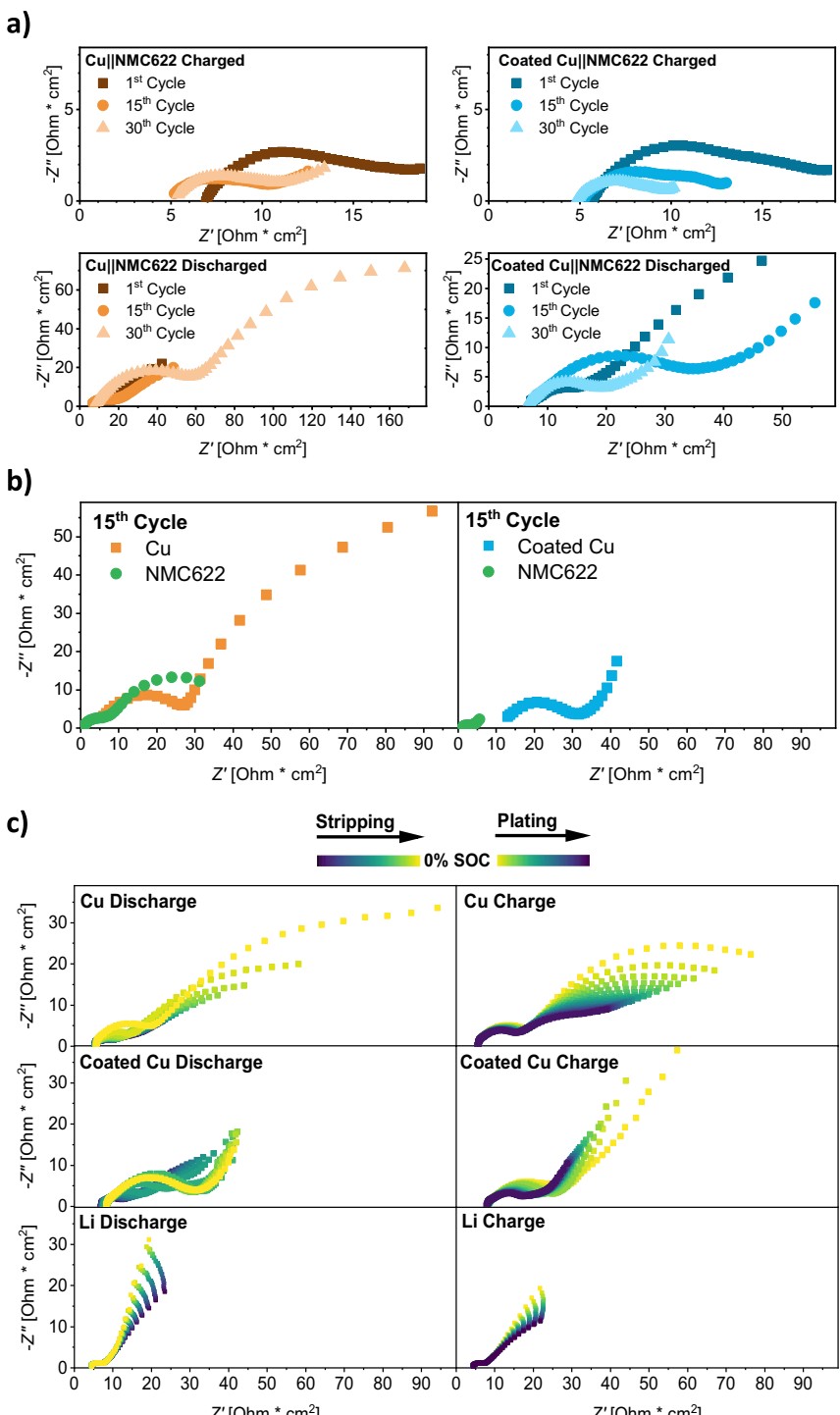

**Fig. 5 | Impedance of Cu‖NMC622 and coated Cu‖NMC622 cells.**
**a** Representative Nyquist plots in the charged and discharged state of Cu‖NMC622 and coated Cu‖NMC622 NMR pouch cells throughout cycling. **b** Representative Nyquist plots of individual electrode impedance data after the 15th discharge cycle using Cu‖NMC622 and coated Cu‖NMC622 PAT three-electrode cells. **c** Representative, state of charge (SOC) dependent Nyquist plots of the last and first 20 dynamic impedance spectra for the 30th and 31st cycle of NMC-based batteries using Cu, coated Cu and Li as the negative electrode, respectively. Impedance data is displayed in a frequency range from 30 kHz to 1.0 Hz with an alternating current of 0.065 mA cm⁻² superimposed on the charge/discharge current of 0.05 mA cm⁻². Electrolyte: LHCE.

1 Hz in the initial cycle (Supplementary Fig. 17f), a fully-resolved semicircle with a depressed arc can be observed after 30 cycles (Supplementary Fig. 17h). This change in frequency dependence indicates accelerated interfacial transport. Revisiting the allocation of charge capacity (Fig. 3b), we note that the rise in impedance throughout cycling for bare copper coincides with the consumption of excess

lithium metal. To examine, if the sudden increase in cell impedance for cells with bare copper is related to a depletion of electric and ionic conductive lithium metal at the negative electrode, a previously established and validated three-electrode PAT cell setup (Supplementary Fig. 19) was employed[73]. With the annular lithium metal reference ring located around the separator, impedance contributions

from the positive and negative electrodes can be distinguished without altering the diffusion pathways[74,75] (Fig. 5b; Kramers–Kronig validation and Nyquist plots Supplementary Figs. 20 and 21, respectively). Comparing the summed spectra of positive and negative electrodes in two- (Supplementary Fig. 18b) and three-electrode (Supplementary Fig. 21) configuration reveals differences in the magnitudes of the semi-circles between cell setups but similarities in their frequency dependence as well as evolution upon cell cycling. Moreover, EIS spectra in three-electrode configuration also display similar impedance of Cu‖NMC622 and coated Cu‖NMC622 cells after the initial discharge but notable differences upon cycling. Due to an accelerated capacity decay in three-electrode cell formats (Supplementary Fig. 22), the drastic increase in discharged state cell resistances with bare copper can already be observed in the 15th cycle for PAT cells rather than after the 15th cycle for NMR pouch cells.

Evaluating positive and negative electrodes separately reveals two semi-circles for each electrode, which can also be visualized using equivalent circuit fitting (Supplementary Fig. 23). Note that both electrodes displaying grain boundary and interfacial resistances are consistent with our previous assignments of these processes in full cell impedance spectra. Compared to bare copper, coated copper negative electrodes exhibit increased Ohmic resistances but similar frequency-dependent impedances in the initial cycle. For the positive electrode, impedance spectra after the 1st cycle are almost identical. Evaluating the impedance after 15 cycles, the three-electrode setup reveals that the rise in impedance occurs for both processes (grain boundary and interfacial) and both electrodes in Cu‖NMC622 cells (Supplementary Fig. 21). In coated Cu‖NMC622 PAT cells, positive and negative electrode exhibit an increase of the high frequency impedance related to grain boundary processes, but a decrease for interfacial processes after cycling. The previously observed acceleration of interfacial processes, indicated by a change in frequency dependence, is also displayed in three-electrode cells and occurs for both electrodes. Cells with bare copper negative electrodes, on the other hand, display increased interfacial resistances upon cycling. Though these observations are made with positive and negative electrodes in either cell chemistry (Fig. 5b), they are only uncovered in the discharged state and after several cycles. Thus, we suggest that differences between the negative electrodes may only be observed when the current collector itself (i.e. Cu or coated Cu) acts as the negative electrode, meaning electrochemical lithium dissolution directly from the substrate instead from an active or excessive lithium metal reservoir.

To substantiate that the increased interfacial resistance in the discharged state with bare copper negative electrodes is related to the electrochemical dissolution of lithium metal from the current collector itself, dynamic impedance spectroscopy (DEIS) is employed, aiming to determine the on-set of the rise in the impedance. Here, an alternating current is applied in addition to a direct charge or discharge current, enabling operando EIS acquisition. Since the sudden increase in impedance was first observed for NMR pouch cells after the 30th discharge, DEIS experiments are carried out during the end of the 30th discharge and the beginning of the subsequent 31st charge cycle (Fig. 5c, Kramers–Kronig validation Supplementary Fig. 24). To keep changes in the SOH and impedance behavior of the system to a minimum during EIS acquisition[76,77], the charge and discharge current densities were reduced to 0.05 mA cm$^{-2}$. In agreement with previous EIS measurements in other cell formats, Cu‖NMC622 and coated Cu‖NMC622 cells exhibit two semi-circles related to grain boundary and interfacial processes in the applied frequency range, which can also be visualized by equivalent circuit fitting (Supplementary Fig. 25). Furthermore, aged Cu‖NMC622 cells also exhibit increased impedance in the discharged state. Here, DEIS reveals a steep rise in grain boundary as well as interfacial impedance within the last 5 spectra (stripping of 0.04 mAh, i.e. 1%) prior to reaching the set cut-off voltage (Fig. 5c). Analogous to its sudden increase towards the end-of-discharge, a fast

decay of impedance is observed upon initial deposition of lithium metal in the subsequent charge process. For cells with coated copper, an SOC-dependent increase in impedance is only observed for the high-frequency grain boundary resistances. Regarding the interfacial processes, a comparatively low resistance is maintained, even upon complete lithium metal dissolution. DEIS spectra obtained upon discharge and subsequent charge of conventional LMBs using excessive lithium metal at the negative electrode demonstrate no SOC dependence for the two semicircles resolved in this frequency range. Here, an increase in the arc of the subsequent semicircle, representing the charge-transfer resistance, is perceivable but not fully resolved in the displayed frequency range (Supplementary Fig. 25g–i). Taking all impedance data from the considered cell formats into account, we suggest the following interpretation. Differences in the impedance of cells containing bare and coated copper only become observable in the discharged state and after consumption of the excess lithium metal reservoir at the negative electrode. As revealed by DEIS, the increased grain boundary and interfacial resistances in Cu‖NMC622 cells appear towards the end of discharge, where lithium is dissolved from the bare copper rather than from a previous lithium layer. In cells containing coated copper negative electrodes, grain boundary and contact resistances also increase upon cycling, but interfacial processes exhibit decay in resistance and characteristic frequencies, indicative of facilitated ion transport. Though these differences between bare and coated copper are observed for the positive and negative electrode, the presence of excessive metal at the negative electrode has a significant impact on their occurrence. Especially the rise of impedance towards the very end of lithium metal dissolution for cells with bare copper negative electrodes and the absence of such effects in conventional LMBs renders these phenomena specific for anode-free cell configurations. Thus, we consider the interfacial transport of Cu‖NMC622 and coated Cu‖NMC622 cells towards the end of electrochemical lithium dissolution as an indicator for the ability to access the lower layer of lithium from the negative electrode itself. Apparently, the enhanced interfacial properties observed with coated copper correlate with less dead lithium deposits.

## Structural analysis of alloying interlayer

With coated Cu‖NMC622 cells displaying improved interfacial transport towards the end of electrochemical lithium metal dissolution, the structural evolution of the functional coating upon charge of the battery requires further characterization. Electrochemically, the negative electrode potential profile of coated copper (Supplementary Fig. 9) indicates alloy formation with a low capacity of 0.05 mAh cm$^{-2}$ due to the thin nature of the functional layers (2–4 μm SrF$_2$-PVDF-HFP and <1 μm Sn@Cu, Supplementary Fig. 12). Given the limited amount of these alloys, no corresponding signal can be identified in the regular $^7$Li NMR spectra intended to quantify metallic lithium. Nevertheless, using modified acquisition and processing parameters, alloy formation can be inferred from a broad and low-intensity signal around 100 ppm (Supplementary Fig. 26). According to previous reports, $^7$Li signals around 100 ppm might originate from the intermetallic compound Li$_{22}$Sn$_5$[78], the Li–Sn alloy with the highest degree of lithiation. However, due to the low signal-to-noise ratio and peak magnitude, an improved spectral resolution is required. Thus, additional $^7$Li NMR experiments at a higher magnetic field strength (14.1 T, $\omega^0_{1H}$ = 600 MHz) are carried out to characterize the formed alloys. Due to intimate contacts of copper and the Sn-layer achieved by redox reactions instead of solution casting[52], scraping powder from the negative electrode to conduct magic angle spinning (MAS) NMR experiments is not feasible for all compounds of the coating. Thus, the negative electrodes were analyzed under static conditions. To further boost signal intensity, the amount of lithium nuclei in the alloy are increased by simultaneously examining two coated copper negative electrodes, each harvested from coin cells after initial charge for two hours (Fig. 6a) and vacuum

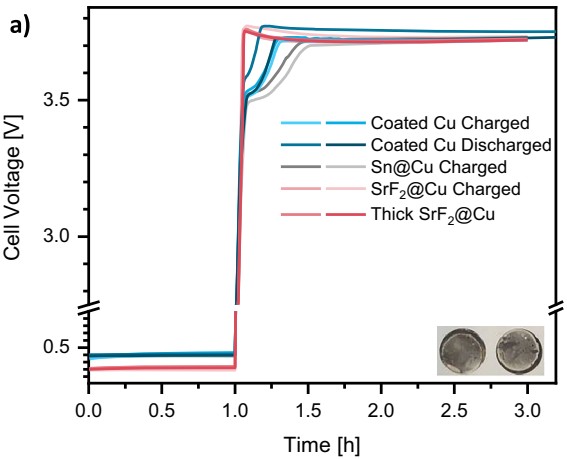

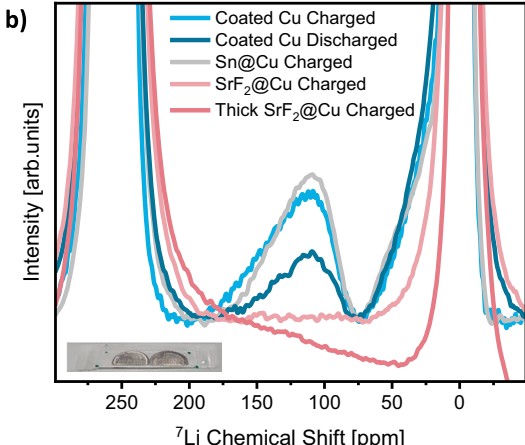

**Fig. 6 | Ex situ $^7$Li NMR experiments to characterize lithium alloy formation.** **a** Voltage profile and exemplary image of negative electrodes harvested from two coated Cu||NMC622, SrF$_2$@Cu||NMC622 and Sn@Cu||NMC622 coin cells after charge for 2 h at 0.2 mA cm$^{-2}$ or the initial cycle, respectively. Slightly lighter colors indicate a repeated measurement with a nominally identical cell. **b** Exemplary image and magnified static ex situ $^7$Li NMR spectra for negative electrodes described in (**a**). The NMR spectra have been subjected to a phase correction of 0$^{th}$ and 1$^{st}$ order to identify the lithium alloy signal.

sealed in an NMR pouch foil to conduct ex situ measurements (Fig. 6b). Owing to these adaptions, the previously observed signal indicative of a Li−Sn alloy phase is captured with enhanced resolution. Still, application of 0$^{th}$ and 1$^{st}$ order phase corrections and subsequent baseline corrections are required to obtain solely positive absorptive signals. Since this processing impacts the chemical shift of peaks, unambiguous assignment of exact alloy stoichiometry remains challenging. To reduce the impact of the processing procedure on chemical shift, the identical data is also Fourier-transformed in magnitude mode (Supplementary Fig. 27), yielding solely absorptive signals at the expense of phase information. Both processing procedures reveal $^7$Li signals around 100 ppm, indicative of the fully lithiated Li$_{22}$Sn$_5$ phases[78]. Aiming to substantiate this assignment, the individual constituents of coated copper, namely Sn@Cu and SrF$_2$@Cu, are characterized separately via $^7$Li NMR spectroscopy. Here, cells containing Sn@Cu and coated copper negative electrodes display a similar initial voltage plateau (Fig. 6a). Fittingly, ex situ $^7$Li NMR spectra of both negative electrodes display an almost identical signal, corroborating its identification as a Li−Sn alloy (Fig. 6b and Supplementary Fig. 27). Since no alloying voltage plateau and no additional NMR signals are observed in the case of SrF$_2$@Cu negative electrodes, even when increasing the coating thickness, alloy formation with coated copper negative electrodes is mainly attributed to the formation of the stoichiometric Li$_{22}$Sn$_5$ phase.

With Li−Sn alloying upon initial charge of coated copper negative electrodes confirmed, questions remain about its evolution and stability throughout charge-discharge cycles. Judging from the potential profile of the negative electrode in three-electrode cells, Li−Sn alloy formation is irreversible during constant current battery operation. While a distinct plateau for alloying is observed upon the initial charge (Supplementary Fig. 9), solely one plateau for lithium dissolution with increasing overpotential towards the end of discharge is observed in all considered cycles (Supplementary Fig. 28). To validate this spectroscopically, ex situ $^7$Li NMR experiments were also conducted on coated copper negative electrodes after the initial discharge in coin cells. Here, a lithium signal of similar chemical shift, indicative of the Li$_{22}$Sn$_5$ phase, but decreased intensity is observed (Fig. 6b). However, a quantitative interpretation of these spectra is not possible, since the processing required to receive absorptive signals distorts the baseline. Processing the spectra in magnitude mode also shows an identical chemical shift but lower intensity of Li−Sn alloys (Supplementary Fig. 27). This can be attributed to a notably decreased alloy capacity for one of the negative electrodes, displayed in the voltage profiles of

coated copper cells. To characterize the formation of Li−Sn alloy phases in more detail, operando $^7$Li NMR experiments with a Sn@Cu|| NMC622 NMR pouch cell are conducted. Since a reduced time for operando NMR experiments (Fig. 7, I – VII) limits the acquisition parameters, battery operation is interrupted every two hours to acquire a spectrum with increased recycle delay, also capturing compounds with slower relaxation (Fig. 7A–G). The first set of operando $^7$Li NMR spectra (Fig. 7I) confirm formation of the Li−Sn alloy as one of the initial processes upon charge. Due to the thin nature of the Sn layer and the resulting limited capacity and spectral resolution, no intermediate Li−Sn phases and solely the peak previously assigned to the Li$_{22}$Sn$_5$ phase are observed. Growth of the metallic lithium signal is only observed after the intensity of the Li−Sn alloy signal reaches its maximum, agreeing with the negative electrode potential profile (Supplementary Fig. 9). Upon further charging and deposition of lithium metal, the Li−Sn signal moves towards lower chemical shifts and its intensity decays (Fig. 7).

Based on our calibration of NMC positive electrode impact on the chemical shift of lithium metal (Fig. 1), the state of charge dependent decay of NMC paramagnetism can contribute to the decreasing chemical shift of the Li-Sn alloy, but should be of a lower magnitude than observed here. The additional variation in chemical shift over time could also be due to the changes in phase correction and processing that are caused by lithium deposition and the corresponding increase in lithium metal signal intensity. Since the theoretical skin depth of $^7$Li RF pulses reduces to 8.4 μm for lithium metal at a higher magnetic field of 14.1 T[32], the continuous deposition of lithium metal can also explain the decay of the Li−Sn alloy signal intensity over time. With the charge capacity increasing by 0.4 mAh cm$^{-2}$ between every in situ spectrum, skin depth limitations may occur from Fig. 7D onwards, assuming dense lithium deposits (5 μm thickness per mAh cm$^{-2}$). Though a certain porosity of lithium deposits might prolong the experimental time until the Li−Sn alloys are buried by lithium metal and the corresponding nuclei are not excited by RF pulses any longer, skin depth effects will limit their detection via $^7$Li NMR at some point upon charge.

Taking all $^7$Li NMR experiments into account, the alloying plateau observed electrochemically can be attributed to the formation of a Li$_{22}$Sn$_5$ alloy phase, which remains stable throughout cycling. Since improved rate capability and exchange current density are well known and reported for Li−Sn alloys[79,80], we consider this phase to be fundamental for the enhanced interfacial properties observed in EIS data. However, an improved lithium nucleation layer morphology on coated copper (Supplementary Figs. 13 and 14) as well as limited SEI

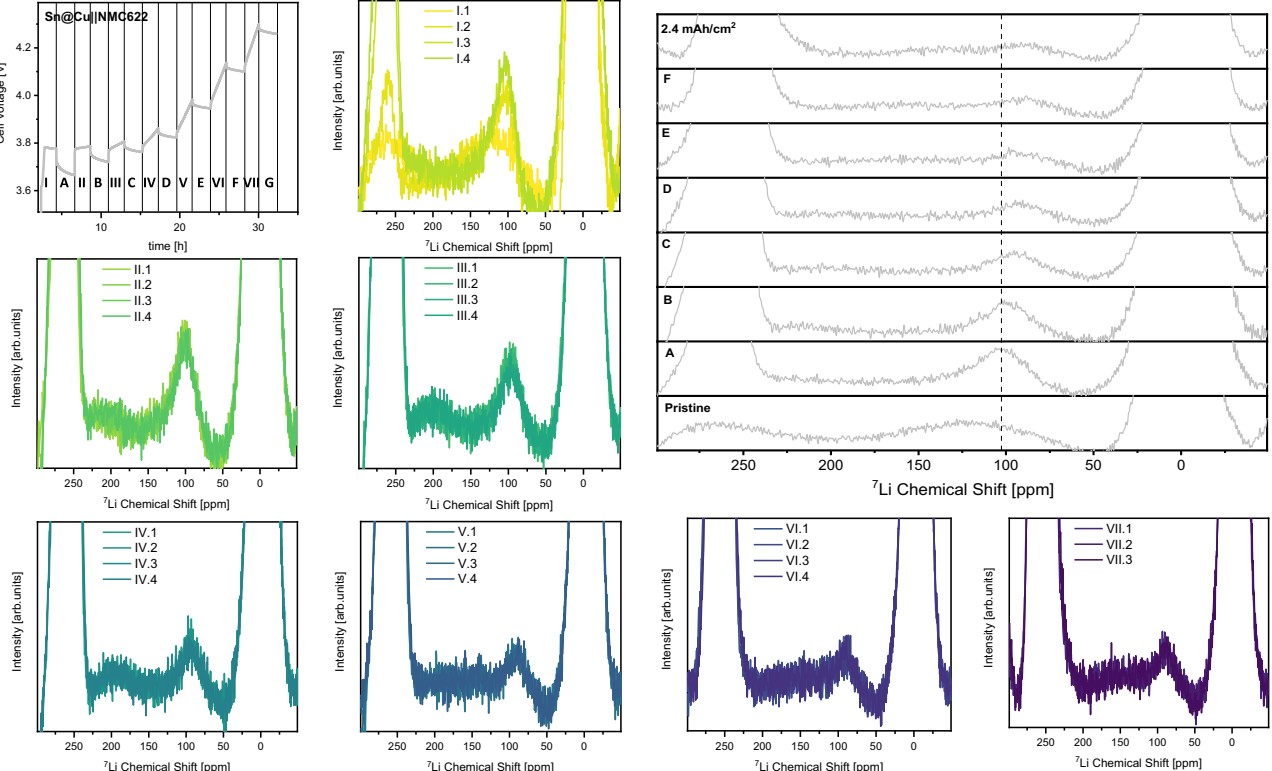

**Fig. 7 | Evolution of the Li–Sn alloy ⁷Li NMR signal upon charge.** Voltage profile of the Sn@Cu||NMC622 NMR pouch cell with constant current operation and operando (**I–VII**) ⁷Li NMR spectra and intermediate resting periods for acquisition of in situ (**A–G**) ⁷Li NMR spectra during charge. The acquisition parameters of an operando ⁷Li NMR experiments yield an acquisition time of 30 min, over which all dynamic processes are averaged. This results in acquisition of four spectra per polarization period. The longer recycle delay of an in situ spectrum at open circuit voltage enables to also detect slowly relaxing species but increases the acquisition time to 2 h and 19 min.

constituent accumulation (Fig. 3) might also contribute to enhanced interfacial transport. Though alloying of lithium and the SrF₂ coating is not observed electrochemically or spectroscopically, this compound still plays a critical role, since an enhanced reversibility of lithium inventory is only observed when combining SrF₂ and Sn in coated copper[52] (Supplementary Fig. 29). In agreement with a previous report on metal fluorides combined with a protective polymer layer, we suggest its beneficial impact to be related with lower SEI dissolution and thereby capacity losses to its re-formation upon cycling[53], which are also quantified in this case (Fig. 3). Optical characterization of the functional layer by ex situ SEM-EDX mapping shows that lithium deposition occurs on top of the functional layer, with Sn and Sr remaining between the metallic lithium deposits and the copper current collector (Supplementary Fig. 30), even after 30 cycles (Supplementary Fig. 31). Nevertheless, lithium nucleation underneath the functional layer was also observed occasionally (Supplementary Fig. 32), indicating better structural integrity and homogeneity as possible factors to actually unlock the full potential of this alloying functional layer.

We conclude that the cells containing coated copper exhibit accelerated and less resistive interfacial transport towards the end of discharge. Next to an improved morphology of the lithium metal nucleation layer and lower SEI-accumulation rate, this can be attributed to the irreversible formation of a $Li_{22}Sn_5$ alloy phase. Since the alloying layer remains between the deposited metallic lithium and the current collector, these differences are observed solely towards the end of discharge, where lithium is dissolved from the alloying interlayer rather than from other lithium deposits. In the reference system, increased resistances towards the end of discharge reflect the challenges associated with completely dissolving lithium from bare copper, which results in higher amounts of electronically insulated lithium deposits.

## Discussion

We herein presented an advanced ⁷Li NMR spectroscopy protocol to distinguish and quantify capacity losses in application oriented and representative anode-free lithium metal pouch cells. The as-developed methodology was applied to unravel the limiting processes in anode-free lithium metal batteries operated with a localized high concentrated electrolyte. Furthermore, the positive impact of a multifunctional coating comprised of Sn, SrF₂ and PVDF-HFP on the reversibility of lithium inventory was determined. While the differentiation of capacity losses revealed continuous interphase formation as the bottleneck in terms of longevity with both, bare and coated copper negative electrodes, accelerated stabilization of the interphases and considerably reduced amounts of electronically insulated lithium deposits were observed for cells using coated copper. Contrary to dead lithium formation caused by fracture of high surface area lithium deposits, the diminished dead lithium accumulation with coated copper negative electrodes did not correlate with the initial lithium metal deposit morphology or its evolution upon cycling (Fig. 8). Employing localized high-concentrated electrolytes, predominantly compact lithium metal species are deposited on both negative electrodes, even though a nucleation overpotential was observed upon initial lithium deposition on bare copper.

While the differences in lithium nucleation overpotential translate into the growth regime of lithium deposition in conventional carbonate electrolytes, heterogeneity differences in lithium nucleation do not result in global morphological differences for lithium deposition with the LHCE (Fig. 9). Rather than avoiding high surface area lithium

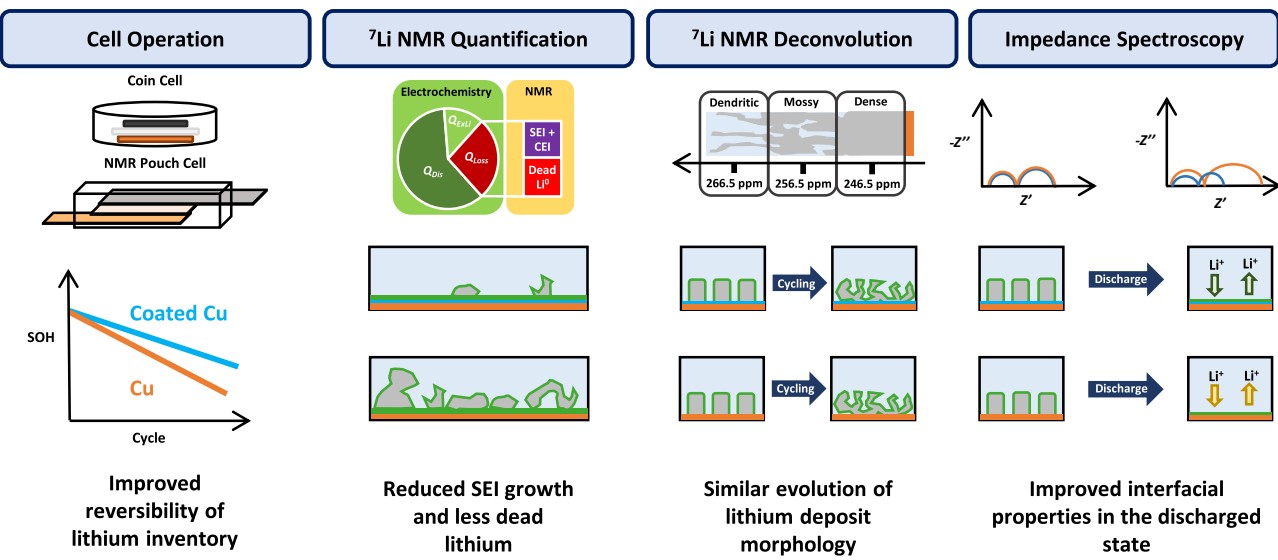

**Fig. 8 | Schematic depiction of the utilized methods and their main findings.** The methodic approaches and experiments are depicted in the top row, while the corresponding observations in NMC-based cells using coated and bare copper negative electrodes are compared in the middle and bottom row, respectively.

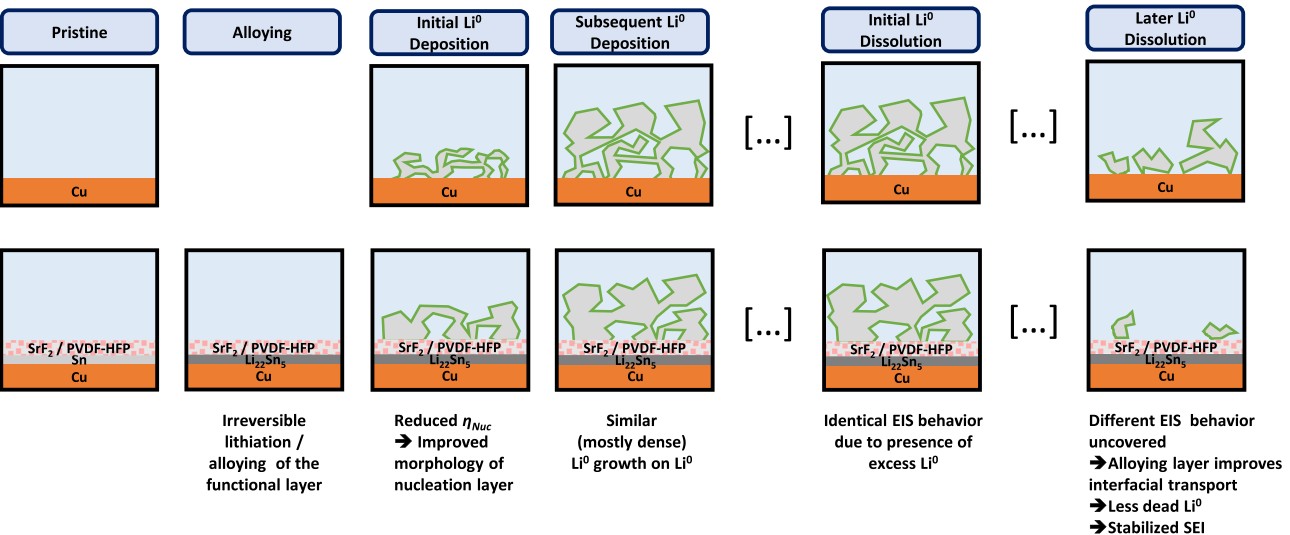

**Fig. 9 | Schematic representation of the impact of the alloying functional layer upon lithium deposition and dissolution.** Processes at bare copper (top row) and coated copper (bottom row) negative electrodes are compared throughout the initial charge and discharge in sequential order, meaning increasing time from left to right.

deposits, the diminished formation of dead lithium with coated copper is bestowed by enhanced interfacial transport towards the end of lithium dissolution. The irreversibly formed $Li_{22}Sn_5$ alloy phase and the $SrF_2$ layer remain between the copper current collector and lithium deposits, accelerating ionic conduction and maintaining electronic and ionic contact in the absence of lithium metal. Here, further investigations, e.g. operando microscopy or X-ray imaging, are encouraged to visualize and validate our understanding of the proposed mechanism of improved dissolution of lithium metal deposits in the presence of alloying coatings.

The methodic approaches presented herein may serve as guidelines for accurate evaluation of capacity losses, reversibility of lithium inventory and morphology of metallic deposits in application oriented anode-free metal batteries via NMR spectroscopy, also expanding the established dead lithium protocol to sodium- or other metal-based systems. To foster future development of materials that boost the reversibility of lithium inventory in anode-free cell concepts, impedance spectroscopy upon complete dissolution of all metallic deposits

was herein established. While a detailed quantification of capacity losses via $^7Li$ NMR spectroscopy requires advanced equipment and design of compatible cells and experiments, impedance spectroscopy serves as an accessible tool to unravel the transport properties for lithium dissolution on the current collector and thereby estimate the materials' ability to diminish the formation of electronically isolated deposits.

## Methods
### Materials
$LiNi_{0.6}Mn_{0.2}Co_{0.2}O_2$ (NMC622) composite electrodes were cast onto aluminum foil (Nippon) using a coating machine (HSCM-20802i, Hohsen Corp.) with a wet film thickness set to 59 μm for one side. The electrode paste resulted from mixing 95 wt.% NMC622 (BASF), 3 wt.% Polyvinylidene fluoride (PVdF1100, Kureha) and 2 wt.% conductive carbon (SuperC65, TIMCAL/Imerys) in 2-N-Methyl-Pyrrolidin ($C_5H_9NO$, Sigma-Aldrich, 99%) in a planetary mixer (Eirich), using a solid to liquid weight ratio of 77:23. After drying at 100 °C and subsequently at 120 °C

overnight, the resulting electrodes (mass loading of 13.5 mg cm$^{-2}$, areal capacity of 2.3 mAh cm$^{-2}$) were calendered (GKL 400, Saueressig Group) to a thickness of 57 μm and a porosity of 30%.

LiN$_{0.5}$Mn$_{1.5}$O$_4$ (LNMO) electrodes were cast onto aluminum foil (Nippon) using a coating machine (HSCM-20802i, Hohsen Corp.) with a wet film thickness set to 88 μm for one side. The electrode paste resulted from mixing 95 wt.% LNMO, 3 wt.% Polyvinylidene fluoride (PVdF5130, Solef/Solvay) and 2 wt.% SuperC65 (TIMCAL/Imerys) with 2-N-Methyl-Pyrrolidin (C$_5$H$_9$NO, Sigma-Aldrich, 99%) with a dispermat (Dispermat CN F2, VMA-Getzmann GmbH), using a solid to liquid weight ratio of 69:31. After drying at 100 °C and subsequently at 120 °C overnight, the resulting electrodes (mass loading of 16.5 mg cm$^{-2}$, areal capacity of 2.3 mAh cm$^{-2}$) were calendered (GKL 400, Saueressig Group) to a thickness of 79 μm and a porosity of 40%.

LiFePO$_4$ (LFP) composite electrodes were casted onto aluminum foil (Nippon) using a doctor-blade technique (ZUA 2000 Universal Applicator, ZEHNTER GmbH) and the automatic film applicator 1133 N (Sheen Instruments). The wet film thickness was set to 275 μm, and a speed of 50 mm/s was applied. The electrode paste resulted from a mixture of 92.5 wt.% LFP (Tatung), 1.5 wt.% Polyvinylidene fluoride (PVdF, Solef 5130, Solvay) and 6 wt.% conductive carbon (SuperC65, Timcal/Imerys) in 2-N-Methyl-Pyrrolidin (C$_5$H$_9$NO, Sigma-Aldrich, 99%) with a solid to liquid weight ratio of 55:45. PVDF was dissolved in 80% of the total 2-N-Methyl-Pyrrolidin amount by stirring overnight. After the addition of LFP and C65, the paste was homogenized in a planetary mixer (Thinky ARM-310, Thinky Corporation) at 1700 rpm for 10 minutes. The identical mixing procedure was repeated after the addition of the remaining 20 wt.% 2-N-Methylpyrrolidin. LFP Sheets were dried at 70 °C overnight, resulting in a mass loading of 14.7 mg cm$^{-2}$ and an aerial capacity of 2.3 mAh cm$^{-2}$. Prior to punching out electrode discs, the sheets were calendered (CLP 2025, Hohsen Corporation) to a thickness of 96 μm and a porosity of 35%.

A copper sheet (Schlenk Metallfolien GmbH, 10 μm) was immersed in glacial acetic acid (Sigma-Aldrich, 99.9%) for 10 minutes. After removing the copper sheet from the acetic acid bath, the remaining acetic acid was evaporated homogeneously by an Argon gas stream in a dry room atmosphere (dewpoint −55 to −60 °C).

Sn@Cu electrodes were synthesized by a dip-coating redox reaction reported previously[52]. Prior to the dip-coating, a piece of copper foil (UBIQ technology Co. Ltd.) was immersed for 10 minutes in 1 mol L$^{-1}$ hydrochloric acid (HCl, Sigma-Aldrich, 37 wt.%) and cleaned with de-ionized water and acetone three times each afterwards. The as-prepared pieces of copper were immersed in a bath of an aqueous solution containing 0.032 mol L$^{-1}$ Tin(II)chloride (SnCl$_2$, Sigma-Aldrich, 98%), 0.591 mol L$^{-1}$ Thiourea (CH$_4$N$_2$S, Sigma-Aldrich, 99%), 0.014 mol L$^{-1}$ Cetrimonium bromide (C$_{19}$H$_{42}$BrN, Sigma-Aldrich, 99%) and 0.186 mol L$^{-1}$ sulfuric acid (H$_2$SO$_4$, Sigma-Aldrich, 98%) for 50 s. After being cleaned with de-ionized water and acetone three times each, the Sn@Cu foils were dried at 70 °C under reduced pressure (<5 ×10$^{-2}$ mbar) for 12 h. To obtain SrF$_2$@Sn@Cu electrodes, referred to as coated Cu electrodes throughout the manuscript, the as-prepared Sn@Cu electrodes were coated with a slurry of SrF$_2$ particles and poly(vinylidene fluoride)-co-hexafluoropropylene (PVDF-HFP, Sigma-Aldrich, ≥98%,) in 2-N-Methyl-Pyrrolidin (C$_5$H$_9$NO, Sigma-Aldrich, 99%). A solid to liquid weight ratio of 1:5 was used, whereas the solid fraction of the slurry was a 4:1 mixture of SrF$_2$ and PVDF-HFP by weight. After mechanically stirring the slurry at 400 rpm for 5 h using a dispermat (Dispermat CN F2, VMA-Getzmann GmbH), the doctor blading technique with a wet gap thickness of 50 μm was used to coat the Sn@Cu foil. The coated copper electrode was dried at 70 °C in a vacuum oven (<5 × 10$^{-2}$ mbar) for 12 h. The SrF$_2$ particles used for coated copper electrodes were synthesized according to a previous report[51]. Sodium citrate (Na$_3$C$_6$H$_5$O$_7$, Sigma-Aldrich, ≥98%,) was added to a 0.05 mol L$^{-1}$ aqueous strontium nitrate (Sr(NO$_3$)$_2$, Alfa Aesar, 99%) solution to achieve a concentration of 0.004 mol L$^{-1}$. After vigorously

stirring for 20 min, an aqueous solution of Sodiumtetrafluoroborate (NaBF$_4$, VETEC, 98%) was added dropwise to achieve a concentration of 0.133 mol L$^{-1}$. This solution was subjected to a hydrothermal oven (2.45 GHz, MARS Microwave 1800 W) and heated at a frequency of 2.45 GHz and power of 300 W, reaching a temperature of 150 °C and pressure of 0.6 MPa. After 10 min, the sample was cooled to ambient temperature, washed with de-ionized water and ethanol three times using a centrifuge and dried at 80 °C in a vacuum oven (<5 × 10$^{-2}$ mbar) for 12 h. To obtain thin and thick SrF$_2$@Cu electrodes, the above-mentioned slurry was cast onto bare copper foil (UBIQ technology Co. Ltd.) at a wet coating thickness of 50 μm and 300 μm, respectively.

The LHCE electrolyte was prepared by mixing lithium bis(-fluorosulfonyl)imide (LiFSI, Elyte Innovations, battery grade), 1,2-dimethoxyethane (DME, Sigma-Aldrich, anhydrous, 99.5%) and tetra-fluoroethyl-2,2,3,3-tetrafluoropropylether (TTE, SynQuest Laboratories, 97%) in a molar ratio of 1:1.2:3. The carbonate-based electrolyte was prepared by dissolving 1 M LiPF$_6$ salt (Elyte Innovations, battery grade) in ethylene carbonate (EC, Elyte Innovations, battery grade) and diethyl carbonate (DEC, Elyte Innovations, battery grade) in a ratio of 3:7 wt. by wt. with an addition of 5 wt.% fluoroethylene carbonate (FEC, Elyte Innovations, battery grade).

## Cells

For two electrode coin cell assembly, a 15 mm negative electrode, 16 mm separator (Celgard2500) wetted with 25 μL of electrolyte, and a 14 mm positive electrode were sandwiched between a 0.5 mm thick stainless-steel spacer with a diameter of 15.5 mm and a 1 mm aluminum coated stainless steel spacer with a diameter of 15.5 mm (MTI). A wave spring (MTI) with 14.5 mm diameter and 1.2 mm height was placed below the stainless-steel spacer. Aluminum-coated coin cell caps (MTI) and a 19 mm diameter piece of aluminum foil were used at the positive electrode to avoid stainless steel corrosion originating from the LiFSI-based electrolyte (Supplementary Fig. 6). The cells were assembled in an Argon-filled glovebox (MBraun, <0.5 ppm O$_2$, <0.5 ppm H$_2$O) and sealed using an automated electric crimper (Hohsen Corporation).

For the $^7$Li NMR investigations, a 0.6 cm × 4.5 cm negative electrode, 1 cm × 4 cm separator (Celgard2500), and a 0.5 cm × 4.0 cm positive electrode were stacked in a PE/PP cell housing under dry room atmosphere (dewpoint −55 to −60 °C) to yield a cell stack with an active area of 0.5 cm × 2 cm (Supplementary Fig. 6). Excluding the top part of the pouch cell to enable injection of the electrolyte at a later stage, the outer parts of the cell housing were sealed with an impulse sealer (Audion Sealboy Magneta). The assembled cells were dried overnight under reduced pressure (<10$^{-2}$ mbar, 45 °C) and transferred to an Argon-filled glovebox (MBraun, <0.5 ppm O$_2$, <0.5 ppm H$_2$O). Here, 50 μL of electrolyte was injected into the cell stack. Afterwards, the PE/PP cell housing was vacuum sealed using an Audionvac VMS43 sealer (Audion). During cycling, a pressure of 13.5 ± 2.0 bar was applied to the pouch cells by external aluminum plates fixated with clamps (Supplementary Fig. 33).

For the three-electrode investigations, commercially available PAT-Cells (EL-CELL) were utilized, comprising an 18 mm negative electrode, 18 mm Celgard2500 separator wetted with 100 μL of electrolyte, and an 18 mm positive electrode (Supplementary Fig. 19). A coated lithium ring, which surrounds the separator and is built into the plastic housing of the cell stack, was used as the reference electrode. The cell stack was sandwiched between a 25 mm-thick stainless-steel lower plunger and an aluminum upper plunger. Cells were assembled in an Argon-filled glovebox (MBraun, <0.5 ppm O$_2$, <0.5 ppm H$_2$O).

The lithium nucleation overpotential was determined in three-electrode T-cells (Swagelok, custom in-house cell design) using 12 mm bare or coated copper as the counter electrode and lithium metal as the working (12 mm diameter, 20 μm thick Li$^0$ on 10 μm thick Cu, Honjo Chemical Corporation) and reference (8 mm diameter, 300 μm thick Li$^0$, Honjo Chemical Corporation) electrodes. The body of the

T-cells was previously covered with an electronically insulating Mylar foil (polyethylene terephthalate, PET). A 12 mm Celgard 2500 PE/PP foil was used to separate the working and counter electrode, while three-layered poly-olefin 8 mm Freudenberg FS2190 disks were employed as a separator for the reference electrode (Supplementary Fig. 19). An electrolyte volume of 30 μL and 90 μL was used to wet the former and latter separator.

All electrodes described herein have been dried in a Büchi-B 585 glass oven under reduced pressure ($<5 \times 10^{-2}$ mbar) for 12 hours at either 120 °C (positive electrodes), 100 °C (bare copper) or 70 °C (coated copper).

## Charge/discharge cycling

Coin and pouch cells were cycled using a MACCOR Series 4000 (Maccor Incorporation) at a temperature of 20 °C. After remaining at open circuit voltage for 12 h (resting), cells were charged with a current density of 0.2 mA cm$^{-2}$ for two cycles and with 0.4 mA cm$^{-2}$ thereafter. The discharge step was carried out at 0.6 mA cm$^{-2}$ for all cycles. The voltage range was set to 2.0–3.8 V, 3.0–4.3 V, and 3.0–4.95 V for LFP-, NMC-, and LMNO-based cells, respectively. For $^7$Li NMR investigations, all the cells were first disconnected from the MACCOR for measurements in the charged state. Afterwards, a constant current discharge and subsequent $^7$Li NMR measurements were carried out. Finally, a constant voltage discharge at the lower cut-off voltage and a temperature of 40 °C was carried out until the current was below 1 μA to remove excess lithium, followed by the final NMR measurement.

Three-electrode cells were cycled using a VMP-3e multichannel potentiostat (BioLogic) with a procedure identical to the procedure invoked in the case of two electrode cells. To obtain corresponding lithium nucleation overpotentials, a fixed current of 0.2 mA cm$^{-2}$ was applied.

## $^7$Li NMR spectroscopy

Static $^7$Li NMR measurements for the quantification of capacity losses and morphology of lithium deposits were performed on a 4.7 T (200 MHz) Bruker Avance III spectrometer using a custom-made probe (Supplementary Fig. 33) suitable for pouch cell investigations. To exclude effects from varying orientation of pouch cells to the magnetic field, cells were fixated using clamps. Prior to the NMR measurements of the actual samples, the $^7$Li NMR chemical shift scale was referenced to 0 ppm using a 1 mol L$^{-1}$ LiCL + 1 g L$^{-1}$ CuSO$_4$ in water standard. 1024 scans separated by a recycle delay of 0.5 s were averaged using pre-saturation and a for each cell individually optimized π/2 pulse with a length of 6.5–7.5 μs (reflecting radio-frequency field strengths of 38.5 and 33.3 kHz) and an amplitude of 80 W. The FID was recorded with a dwell time of 0.1 μs and a time domain of 32k. To quantify the capacity consumed by dead lithium formation, the capacity per NMR integral was determined by dividing the discharge capacity by the integral difference between the charged and discharged states. The integral for lithium metal after the CV discharge step was multiplied by this conversion value, yielding the capacity lost due to dead lithium formation. The remaining capacity losses were considered to be due to (re)formation of the SEI and CEI layers, which were not further differentiated. All integrals obtained for $^7$Li NMR lithium metal peaks were corrected by a scaling factor depending on the NMC622 cathode state-of-charge upon spectral acquisition. For example, $^7$Li NMR spectra acquired at 0% SOC were multiplied by a factor of 1.08 before comparison to $^7$Li NMR spectra recorded in the charged state of NMC622 in order to take the losses of intensity due to NMC SOC differences into account.

Static $^7$Li NMR experiments for the structural characterization of the alloying interlayer were recorded on a Bruker AVANCE NEO 14.1 T spectrometer equipped with a customized static flat-coil VT NMR probe made for operando NMR experiments of pouch-type batteries. Copper anodes with SrF$_2$, Sn or SrF$_2$@Sn coatings have been investigated ex situ after depositing lithium metal onto the negative

electrodes in coated Cu ‖NMC622 cells for 2 h at 0.2 mA cm$^{-2}$. For each negative electrode, two cells were disassembled in an Argon-filled glovebox (MBraun, O$_2$ <3ppm, H$_2$O < 0.1ppm). To fit into the pouch bag sample compartment, the anodes were cut in half using a scalpel. Two electrode halves with the inactive side back-to-back were placed next to another set of two electrode halves (Fig. 6b), allowing to simultaneously record $^7$Li NMR spectra of both negative electrodes (total surface area of 3.08 cm$^2$). Prior to transferring the samples to the NMR spectrometer, the pouch bags were vacuum sealed inside the glovebox. To avoid $^7$Li NMR signal losses from Li metal or metal-like alloys due to $T_2$-relaxation under static conditions, the pulse program only consisted of a π/2 pulse with a maximum rf-bandwidth of 20 kHz (SFO1 100ppm), followed by a short ring-down delay of 6.5 μs and a dwell time of 0.1 μs. 16k scans and a recycle delay of 0.5 s resulted in a total experimental time of 2 h and 25 min. Phase-offsets have been corrected by 0$^{th}$ and 1$^{st}$ order phase correction and subsequent base-line correction. All the NMR spectra were processed within Bruker TopSpin 4.4.1. Please note that all spectra can only be qualitatively evaluated under these conditions. For combined operando/in situ $^7$Li NMR experiments, a Sn@Cu ‖NMC622 NMR pouch cell was assembled as previously described. The cell was placed inside the RF-coil of the probe and connected to a Biologic SP-150e galvanostat. In agreement with operating rates utilized throughout the manuscript, the cells were charged (lithium deposition) at 0.2 mA cm$^{-2}$ to a cut-off voltage of 4.3 V, which was reached after 13 h and 7 min (2.6 mAh cm$^{-2}$) of Li metal plating. For $^7$Li NMR spectra during operation, the number of scans and the recycle delay were reduced to 4096 scans and 0.4 s, respectively, decreasing the acquisition time to 30 min. Since detection of slow relaxing compounds requires longer recycling delay, in situ $^7$Li NMR experiments were carried out in addition to operando experiments, interrupting the charging process every 2 h. Here, a $^7$Li NMR spectrum was acquired with a recycle delay of 2 s and 4096 scans, resulting in a rest period of 2 h 19 min and 5 s between charging intervals of 2 h. All spectra have been processed equally.

To obtain insights into the lithium morphology, $^7$Li NMR spectra in the charged state were deconvoluted with a customized MatLab script based on peakfit.m by Thomas C. O'Haver[81]. Since the state-of-charge dependent paramagnetism of the NMC cathode active material affects the local magnetic field and hence observable $^7$Li NMR chemical shifts, a calibration series was carried out to determine the state-of-charge corrected $^7$Li NMR chemical shift windows for all the considered lithium deposit morphologies (Supplementary Fig. 3). Here, Cu‖NMC622 NMR pouch cells were charged to varying state-of-charges and disassembled afterwards. The negative electrodes with in situ deposited lithium metal were reassembled against pristine NMC622 electrodes, demonstrating the impact of positive electrode state-of-charge. The NMC622 electrodes with varying state-of-charge were reassembled against a 2 cm × 0.5 cm pristine lithium metal electrode (20 μm Lithium on 10 μm copper), which exhibits a defined chemical shift of ≈242.9 ± 0.1 ppm without any paramagnetic influence of NMC622. Since the NMC622 electrode at 100% state-of-charge alters the actual $^7$Li NMR chemical shift of pristine lithium metal to ≈246.7 ± 0.3 ppm, the chemical shift region for dense lithium metal deposits was defined from 244 ppm to 249 ppm. Previously reported chemical shift regions[30–32] for mossy and dendritic lithium deposits were also adjusted by ≈+3.5 ppm, defining mossy deposits as the chemical shift region from 254 ppm to 259 ppm and dendritic lithium deposits in the chemical shift region from 264 ppm to 269 ppm. All $^7$Li NMR spectra in the charged state were fitted using three Voigt functions that were restrained to the respective chemical shift regions and also limited to the peak width obtained for a pristine lithium metal electrode in the vicinity of an NMC622 electrode at 100% state-of-charge. The respective fractions of each deposit morphology were then calculated by relating the respective integral to the total $^7$Li NMR signal integral.

## Electrochemical impedance spectroscopy

Electrochemical impedance spectroscopy (EIS) on NMR pouch cells was conducted using a Zennium Pro potentiostat (Zahner). An alternating current with an amplitude of $100\,\mu A\,cm^{-2}$ was applied in a frequency range from 300 kHz to 0.1 Hz, averaging 6 times 10 data points per decade. Dynamic EIS (DEIS) acquisition was carried out in CR2032 coin cells using a VMP-3e multichannel potentiostat (BioLogic). Prior to DEIS acquisition, cells were cycled following the regular protocol. Upon DEIS acquisition, the current density upon charge and discharge was reduced to $0.05\,mA\,cm^{-2}$ (ca. 0.025C) and superimposed with an alternating current of $0.065\,mA\,cm^{-2}$ (130% of the DC current) in a frequency range from 300 kHz to 0.1 Hz averaging 6 times 10 data points per decade. PAT-Core Cells (EL-CELL) were operated on a VMP-3e multi-channel potentiostat (BioLogic) to enable three-electrode EIS acquisition. 18 mm diameter positive and negative electrodes were sandwiched between a standard stainless-steel plunger at the negative electrode (250 height number) and an aluminum plunger at the positive electrode to avoid stainless steel corrosion from the LHCE. To maintain comparability in the electrochemical behavior, the insulation sleeves were customized (EL CELL) using the same separator as in other cell formats used in this study (Celgard2500). The annular lithium metal reference ring is enclosed in the insulation sleeve and oriented horizontally in the separator frame but not in between the electrodes, achieving symmetrical field distribution without impacting diffusion pathways.

Prior to data evaluation, the Kramers–Kronig transformation was employed to calculate relative residuals between experimental and transformed data via RelaxIS software (Version 3.0.17 Build 10). Since systematic errors were observable for frequencies below 1 Hz for all impedance data, only reliable data points at frequencies > 1 Hz were evaluated. Equivalent circuit fitting (Supplementary Figs. 17, 23, 24) was also carried out using RelaxIS software (Version 3.0.17 Build 10).

## Determination of error margins

Error margins (e.g. for the quantification of capacity losses or lithium morphology) were estimated by reproducing experiments at least once. The obtained data sets for nominally identical experiments were evaluated separately, yielding their average as the value for the data point and the standard deviation between measurements as the error margin. When possible, identical measurements were both displayed in the respective plots.

## Cross-section

Electrodes used for cross-sections were harvested from cycled CR2032 coin cells in the charged and discharged state after disassembly in an Argon-filled glovebox (MBraun, <0.5 ppm $O_2$, <0.5 ppm $H_2O$). A representative piece of the electrode was cut inside the glovebox with a scalpel and subsequently used for the cross-section. To avoid exposure to ambient conditions, the sample was transferred to the cryo cross-section polisher (IB-19520CCP, JEOL) in a vacuum sealed sample holder. Samples were cut under cryo conditions (liquid $N_2$ cooling, −120 °C) for 2.5 h at 5.0 kV and polished afterwards at 2.0 kV for 1 h. After returning to room temperature, samples were transferred to the glovebox for SEM preparation in a vacuum sealed sample holder.

## SEM-EDX

Scanning electron microscopy images were obtained with a Zeiss Crossbeam 550 electron microscope (Cryo-FIB-SEM, Carl Zeiss Microscopy GmbH). To prevent exposure to ambient conditions, all samples were transferred with a vacuum-sealed sample holder. Top-view and cross-section images were taken at 3 kV accelerating voltage with an in-lens detector at a working distance of 5 μm and an aperture size of 30 μm. The elemental composition of the electrode surface and cross-sections was investigated by energy-dispersive X-ray spectroscopy (EDX) with an Ultim Extrem detector (Oxford Instruments). Here, the accelerating voltage was increased to 10 kV. Evaluation of EDX data was performed via Aztec software (Oxford Instruments).

## Data availability

Source data for figures displayed in the main text are provided using figshare under https://doi.org/10.6084/m9.figshare.28597526. [7]Li NMR spectra for calibrating the impact of NMC positive electrodes on the [7]Li chemical shift and integral of metallic lithium are provided using figshare under https://doi.org/10.6084/m9.figshare.27277413. [7]Li NMR spectra for Cu||NMC622 cells are provided using figshare under https://doi.org/10.6084/m9.figshare.27277350. [7]Li NMR spectra for coated Cu||NMC622 cells are provided using figshare under https://doi.org/10.6084/m9.figshare.27277347. [7]Li NMR spectra for the characterization of lithium alloys are provided using figshare under https://doi.org/10.6084/m9.figshare.28595855.

## Code availability

The modified MatLab is based on peakfit.m by Tom O'Haver[81], Copyright (c) 2018, Tom O'Haver. All rights reserved. Redistribution and use in source and binary forms, with or without modification, are permitted provided that the following conditions are met: Redistributions of source code must retain the above copyright notice, this list of conditions and the following disclaimer. Redistributions in binary form must reproduce the above copyright notice, this list of conditions and the following disclaimer in the documentation and/or other materials provided with the distribution This software is provided by the copyright holders and contributors "as is" and any express or implied warranties, including, but not limited to, the implied warranties of merchantability and fitness for a particular purpose are disclaimed. in no event shall the copyright owner or contributors be liable for any direct, indirect, incidental, special, exemplary, or consequential damages (including, but not limited to, procurement of substitute goods or services; loss of use, data, or profits; or business interruption) however caused and on any theory of liability, whether in contract, strict liability, or tort (including negligence or otherwise) arising in any way out of the use of this software, even if advised of thepossibility of such damage. Utilization of the herein presented modified MatLab script with constrained fit parameters is granted under the condition that this reference is properly cited. This script is provided using figshare under https://doi.org/10.6084/m9.figshare.29352173.

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

## Acknowledgements

The authors acknowledge generous support from the German Federal Ministry of Education and Research (BMBF) and the Taiwanese National Science and Technology Council (NSTC) within the framework of the German-Taiwanese research project LiBEST3 (BMBF 13XP0559A/NSTC 112-2639-E-011-ASP). Additionally, NMR experiments regarding the structure of the alloying interlayer were enabled by generous funding from the German Federal Ministry of Education and Research (BMBF) via the grant "For-Analytik" (13XP0445). Furthermore, the authors thank Marian Stan for support with the SEM-EDX measurements as well as Bärbel Tengen for help with the preparation of the sample cross-sections.

## Author contributions

L.W. conceived, designed and performed the experimental studies, except characterization of the alloying interlayer at a field strength of 14.1 T, which was carried out and evaluated by J.H.T. S.K.J. prepared the coated copper negative electrodes. M.M. guided the experimental setup of impedance measurements. L.W. carried out the data analysis and wrote the manuscript. B.J.H. contributed by discussing the results, required methodic approaches and editing the manuscript. G.B. and M.W. supervised the work, guided and edited the writing of the manuscript. G.B. and B.J.H. acquired funding. All authors discussed the results and commented on the paper, agreeing to its submission.

## Funding

## Competing interests

The authors declare no competing interests.
