## [Transparent Peer Review file · Nature Communications]

Origins of lithium inventory reversibility with an alloying functional layer in anode-free lithium metal batteries

Corresponding Author: Dr Gunther Brunklaus

Version 0:

Reviewer comments:

Reviewer #1

(Remarks to the Author)

The manuscript by Wichmann and colleagues outlines an NMR method to quantify capacity loss in anodeless cells. The method is then utilized to determine the mechanism of lithium reversibility on bare Li and coated current collectors. The script presents an interesting methodology and observations, however, it suffers from the limited SEI/interphase structural data to corroborate the findings. For example, the authors postulate that lithium metal embedded in the matrix benefits from reduced SEI layers, some evidence of this from another technique is required before any firm conclusion can be drawn. Currently, this paper is limited by solely presenting an observation.

The authors link enhanced interfacial transport rather than improved morphology of the lithium metal deposits. But there is no evidence to support that the morphology is not a factor in reduced isolated lithium deposits. The paper requires some evidence that the morphology is consistent in both the coated and bare cells.

In the coated cells, the script should outline where the Li is plated?

The script states no ${}^7\text{Li}$ signal corresponding to Sn/Sr alloys is observed. The Sn alloy would appear between 0-100 ppm (J. Phys. Chem. C 2010, 114, 14, 6749–6754). This region of the spectra should be presented in the SI to support this statement.

The ${}^{19}\text{F}$ NMR could also be used to show that the coating is not affected by the insertion of Li.

Some of the figures contain more data than the accompanying key. For example, Fig 4c the key has 6 data points and the figure contains far more than these 6. The same issue is seen in Figure 5. It is difficult to review the images until the reason for these additional data points is given.

What are the effects of the Cu current collector on the NMR signal attenuation?

Quantitative NMR requires knowledge of the recycle delay, this is not presented.

The NMR figures are presented as intensity against shift, this presumes the FWHM remains constant. The integrated area would be more appropriate.

The authors use the terms upfield/downfield, this terminology is only relevant if the authors used a field-swept instrument. These should be changed to lower/higher frequency or in terms of the Knight shift.

How is the reduction in Li metal intensity due to cell disassembly controlled? Can the authors take the NMR of the separator to show the unaccounted for Li?

The errors in the quantitation by NMR, how are they determined?

The authors should fit the Nyquist plots in the SI.

There are statements within the text that require referencing, such as line 105-107.

Many of the reference sources are 'not found', therefore it is difficult to comment on whether the script is referenced appropriately.

Reviewer #2

(Remarks to the Author)

The authors present a detailed in-situ ⁷Li NMR spectroscopic study combined with impedance spectroscopy of an “anode-free” Li metal battery to unravel the distribution of Li in the different battery components, such as cathode, in-situ formed anode, interlayers and the formation of isolated Li deposits, as a function of SOC and SOH.

“Anode-free” Li metal batteries are a very promising battery design for future applications, combining the possibility to boost energy density and at the same time limiting safety hazards and costs by reducing the Li metal content to the volume/mass stored in the positive electrode. As such, the presented study is of high relevance for a better understanding of the mechanisms during Li plating and stripping at the negative current collector and thus for the further development of “anode-free” Li metal batteries.

The authors present comprehensive and high-quality in-situ ⁷Li NMR data on six “anode-free” Li metal battery systems, which used NMC, LMNO or LFP as cathode materials against Cu or coated Cu with an alloying functional layer as negative current collectors. The authors provide a careful analysis of ⁷Li NMR spectra to quantify the Li distribution in different compartments as a function of (dis)charge cycles. Furthermore, the authors investigate the morphology of Li deposits using ⁷Li NMR and SEM. Finally, impedance spectroscopy is used to characterize transport properties and derive a mechanism for dead Li in LHCE.

The authors carefully evaluated the influence of paramagnetism of NMC cathode material as a function of SOC on both ⁷Li chemical shifts as well as signal intensity for quantitative analyses. In this way the authors are able to quantify the amount of dead lithium (electronically isolated Li metal), and the formation of interlayers (CEI and SEI) during cycling of the batteries. Concerning this procedure, I would like to ask the authors to clarify a few minor things:

1. In Fig. 1 signal intensity was used as y axes label, which could either refer to relative height of the signals or their integral (which I guess was meant). Please clarify.
2. Furthermore, in Fig.1 the difference between light and darker colour (e.g. light and dark red, light and dark green) spectra is not explained. Please also clarify this.

The authors then show that ⁷Li NMR in combination with absolute capacity losses can be used to quantify the amounts of kinetically hindered and dead lithium compared to active “strippable” Li as a function of (dis)charge cycles.

3. In Figure 2 a and b) significantly higher chemical shift were observed for Li deposits in both charged and discharged states (between 250 and 280 ppm) compared to the shifts shown in Fig.1 (between 240 and 250 ppm). This currently remains unclear in the manuscript.

The authors proceed with comparing Li plating on Cu vs. a coated Cu with SrF₂+PVDF HFP@Sn@Cu, with the latter clearly improving the reversibility of Li plating/stripping.

4. I wish the authors would have included more information on the coating itself. At the current stage there is very little on the state-of-art of this coating or why it was chosen.
5. Page 12, lines 350 to 360: I propose that the authors refer to the experimental details to clarify how the relative amount of “strippable” Li, dead Li and Li in interphases are calculated and converted to capacity losses. Currently, this is hard to follow.

The authors then show that Cu and coated Cu result in similar Li morphologies starting from dense Li towards mossy and dendritic Li for deposited Li as a function of battery cycling by established ⁷Li chemical shifts for the different morphologies and SEM Images.

6. It could be helpful to indicate dense, mossy and dendritic Li in the SEM Images in Fig. 4a)
7. Is the morphology the reason for the shift difference in Fig.1 and Fig.2 (first dense and later mossy/dendritic Li-compare comment 3.)? This should then be stated somewhere in Fig.2 to avoid confusion.

And finally, the authors state that no Li signal was observed in either Sn or Sr alloys. At least for Li_xSn a very different shift range (between 0 and 100 ppm) is expected (compare J. Phys. Chem. C 2010, 114, 6749–6754). Did the authors vary the offset to test for these shifts as only very low B1 field strength was used in the study? If the authors did, these spectra should be included in the SI. How thin is the coating? Would it be possible to record ¹¹⁹Sn NMR spectra of the coating itself?

Further insight into how the coating itself changes as a function of cycling might help to explain the role of the coating for the improved Li inventory reversibility and might provide directions for future coatings.

It could further strengthen the postulation “We postulate that lithium metal embedded within the alloying polymer matrix benefits from less accumulation of SEI layers on the outside of lithium deposits and an improved interconnection by the alloying metals within the metal deposits, which then translates into comparatively low interfacial resistances in the discharged state and a reduced formation of isolated ‘dead’ lithium deposits (Figure 7).” (page 21) and support the findings from impedance spectroscopy.

Furthermore, I have a few more detailed, mostly minor suggestions/ comments:

8. Please make sure that all abbreviations, such as LFP, LMNO, SOC, etc. are defined upon their first occurrence. This will make it easier for a broad readership to follow the manuscript.
9. Page 14, lines line 411, please provide a reference for "previously defined chemical shift boundaries"
10. Page 14, line 413: was the FWHM of the three voigt functions fixed. If yes, on which bases?
11. Page 16, line 440: please define the abbreviation HSAL.
12. Page 21, line 583: "alloying polymer matrix" seems misleading as Sn or Sr are alloying with Li and not the polymer, which is only the binder. Correct? (more details on the coating would be useful – compare comment 4)

Reviewer #3

(Remarks to the Author)

Manuscript Number: NCOMMS-24-33721

Title: Origins of lithium inventory reversibility with alloying functional layers and a localized high concentrated electrolyte in 'anode-free' lithium metal batteries

Article Type: Research Paper

The manuscript "Origins of lithium inventory reversibility with alloying functional layers and a localized high concentrated electrolyte in 'anode-free' lithium metal batteries" by Wichmann et al. explores the reasons behind improved metal inventory reversibility in 'anode-free' lithium metal batteries with a multi-functional alloying layer on the copper current collector. The study highlights two key findings: Quantitative ^7Li NMR spectroscopy shows that accelerated formation of a stabilized Solid Electrolyte Interphase (SEI) and reduced electronically insulated Li deposits are crucial for enhanced 'anode-free' lithium metal battery longevity. Additionally, the reduced formation of isolated lithium deposits is attributed to improved interfacial transport properties rather than better lithium deposit morphology. Wichmann et al. primarily used two techniques in the study: static NMR spectroscopy and electrochemical impedance spectroscopy (EIS). The authors utilised a two-electrode setup in standard coin-cell and pouch-cell geometries and for the three-electrode setups they employed a PAT-Cells configuration with a coated lithium ring as the reference electrode, as well as a custom three-electrode T-cell (Swagelok) where both the counter and reference electrodes are lithium, and the working electrode is commercial Li₀ on Cu. The study compared two-electrode setup systems using coated (multi-functional alloying layer: Sn, SrF₂ and PVDF-HFP) and uncoated copper sheets.

The conclusions throughout the manuscript are predominantly based on the NMC622 composite electrode chemistry using localized high-concentration electrolytes (LHCE) in 'anode-free' lithium metal battery systems.

In Figure 2b), NMR-based evaluation lacks evidence for other chemistry-related measurements. This raises the question of why excess lithium was not detected in other systems, such as LMNO or LPF composite electrodes (there are references to this in the literature), to which it could potentially be linked? Additionally, the effect of different electrolytes is poorly addressed. Even though the authors mention that each individual cell setup requires separate calibration, there is a lack of supporting evidence and reference experiments performed.

The manuscript neglects fundamental pillars that influence the data interpretation, such as the test cell orientation-effect bulk magnetic susceptibility (BMS) shifts (Figures 1,2), in addition to the ^7Li signals. It is well established that the shape of the sample, the packing of the material, and the sample's orientation relative to B₀ contribute to the variable effects of BMS. Thus, due to the different geometries used, results from a two-electrode system cannot be directly compared/transferred to those from a three-electrode system and vice versa. The same is valid for EIS analysis.

It should also be noted that electrochemical tests have been planned in the two-electrode and three-electrode configurations, which is an excellent start. However, the interpretation of conducted EIS experiments is questionable. The authors do not master this domain and the EIS proofs the lack of understanding of this experimental method and interpretation of its results.

§EIS complex impedance plot must be squared and ortho-normed. It remains unclear why the authors use the electrochemical impedance spectroscopy method in this study. What processes were they hoping to detect in the frequency range 300 kHz - 0.1 Hz?

§Experiments conducted are not performed under steady-state conditions and do not follow the rules of linearity, stability, causal and finite. follow the rules of linearity, stability, causal and finite. A. Lasia / Electrochemical Impedance Spectroscopy and its Applications, doi: 10.1007/978-1-4614-8933-7

§Furthermore, in a two-electrode system, it is very misleading to assume that impedance spectroscopy only separates information for the negative or positive electrode. Even if it is measured in a totally charged or discharged state, and one might just as well assume that the total impedance of a cell is now determined by only one electrode, this is not the case. Batteries are not electric circuits, so the impedance at the complex level of the cell is determined by several factors.

§The position of the reference electrode concerning the working electrode is of great importance. Unfortunately, it is not discussed in the manuscript. Without validating different cell geometries and implementing the reference electrode (RE), including factors related to the chosen material, position, and design of the RE, artefacts can be introduced, as several factors can influence the measured data. Additionally, demonstrating the quality of EIS measurements using Kramers-Kronig

relations is necessary.

“Consistent with the impedance data from NMR pouch cells in a two-electrode configuration, similar DRT spectra are observed in the initial cycles with both negative electrodes.”

Please note, that the DRT method is generally unsuitable for systems with blocking electrodes, such as those encountered in batteries. Even if DRT analysis was used, it is necessary to fully assess the accuracy and resolution of the DRT inversion obtained from the time measurements.

In summary, the EIS experiments and analysis do not support the claim that “the reduced formation of isolated lithium deposits is attributed to improved interfacial transport properties rather than better lithium deposit morphology.”

§What was the contact pressure used for assembling the test cell with various thicknesses of the NMC622, LMNO, LPF vs coated (multi-functional alloying layer: Sn, SrF₂ and PVDF-HFP) and uncoated copper sheets?

§What was the final thickness of the electrode in the re-assembled test-cell? These physical (thickness) properties clearly do not manifest as noticeable differences in electrochemical behaviour when measured in a three-electrode setup, why?

The data are incomplete, and the execution and analysis leave much to be desired.

In summary, the electrochemical measurements are not convincing, and the data analysis does not meet the relevant criteria of Nature Communications. This manuscript still needs careful editing by writers with expertise in electrochemistry.

Version 1:

Reviewer comments:

Reviewer #1

(Remarks to the Author)

The authors have made substantial changes to the manuscript. However, I still find the presented work convoluted and lacking general applicability. The universal applicability of this methodology is very low due to the substantial amount of experimental corrections (orientation, shift, intensity, FWHM, B1) required to achieve results.

The discharge capacities achieved here is lower than what other anode-free architectures present in the literature (Nature Energy, 4, 683–689, 2019). If practical capacities >200 mAh/g is employed how much Li is deposited and would this cause a skin depth issue? Without achieving commercially required discharge capacities (or even improvements on current state of the art NMC systems) then I would question the practicality of this system.

There is a lack of structural analysis of the SEI leading to broad claims to be made in the manuscript i.e. 'The irreversibly formed Sr-Li and Sn-Li alloys' - the authors present no evidence of these alloys. Therefore the mechanism is purely speculative.

The abstract is rather vague and fails to summarize the findings.

The authors should measure T1 rather than the recycle delay.

Figure S5b is severely truncated and should be rerun with a longer acquisition time. Likewise, S10a should be phased appropriately.

'the penetration depth of radiofrequency pulses invoked to excite nuclei of interest is calculated to be 14.7 μm. Thus, even cells containing considerably thin lithium metal electrodes with a thickness of 20 μm cannot be investigated quantitatively.' Are the authors using a flat surface coil? A cylindrical coil would allow 2 x 14.7 μm penetration depth.

In Fig S3 are the authors using a permanent magnet? If so, what is the resolution and have they considered using echoes to reduce inhomogeneities?

Reviewer #2

(Remarks to the Author)

The authors did improve their manuscript “Origins of lithium inventory reversibility with alloying functional layers in ‘anode-free’ lithium metal batteries” and clarified my previous concerns. I recommend publishing after addressing the following comments:

I have a few further remarks:

1. The spectra presented in Fig. S10 are dephased making a judgment of a (broad signal and small) signal for potential alloying to Li_xSn or Li_xSr difficult. It may also be useful to add a baseline at 0 as guide for the eye.

In the manuscript:

Compared to bare copper, the loss of capacity due to interphase formation is higher initially but the ongoing lithium consumption due to interphase reformation throughout cycling occurs at a lower rate. We interpret this behavior as an accelerated formation of interphases with improved durability.

2. To unambiguously differentiate in SEI formation and durability, one would need the ^7Li data for coated Cu after 45 cycles. After 15 and 30 cycles the amount of SEI is similar for Cu and coated Cu within the error ranges. I feel that the statement made is too strong for the current data bases. Additionally, the R^2 value of 99.9 % for the fit of the orange squares (interphase formation on Cu) doesn't seem to be correct considering the outlier after 30 cycles. Please correct.

In the manuscript:

In contrast to the loss of lithium due to interphase formation, a reduced amount of 'dead' lithium formation when using the coated copper is already observable in the initial cycle.

3. Fig. 3d states 0% and 0% dead lithium formation for both Cu and coated Cu. It would be beneficial to add the exact numbers at least in the text, to make the difference obvious. A direct comparison of the ^7Li spectra for coated and uncoated Cu (in the SI) after the 1st cycle could also help in this respect. In Fig. S9 I cannot make out any significant difference – I am not sure whether this is due to the representation of the data, or whether there is no significant difference between the spectra. Please clarify.

In the manuscript:

We conclude that the cells containing coated copper exhibit accelerated and less resistive interfacial transport upon dissolution of the lithium metal deposits in contact with the current collector. Here, irreversible alloy formation provides the functional layer with facile lithium conduction.

4. An alloy formation is one of the main conclusions from the paper, however direct experimental proof is missing. I agree with the authors that low Li contents in addition to low amounts of coating may cause that the amount of formed alloys is below the NMR detection limit. However, as alloying seems to be the main point of the paper, I feel that direct evidence for the formation of the formed alloys is necessary. One could think about a model system, where the material used for the coating is artificially increased, so that at least the formation of alloys can be proven

Reviewer #3

(Remarks to the Author)

Dear Authors

Manuscript Number: NCOMMS-24-33721A

Title: Origins of lithium inventory reversibility with alloying functional layers in 'anode-free' lithium metal batteries

Article Type: Research Paper

The revised manuscript "Origins of lithium inventory reversibility with alloying functional layers and a localized high concentrated electrolyte in 'anode-free' lithium metal batteries" by Wichmann et al. contains changes. Still, unfortunately, the substance of the arguments have not been addressed.

1) "By combining impedance and ^7Li NMR spectroscopy, the reduced formation of isolated lithium deposits is found to be facilitated by enhanced interfacial transport properties rather than a better morphology of metallic lithium deposits."

This claim is still not covered, nor can it be proven by the tests performed by the EIS, so the word impedance should be removed from this sentence in the manuscript.

2)What processes were they hoping to detect in the frequency range 300 kHz - 0.1 Hz?

...Furthermore, the frequency range between 300 kHz and 0.1 Hz is commonly employed in battery related research. Typically, electrolyte resistance, contact resistances, grain boundary as well as interfacial processes are observable. Depending on the cell chemistry and SOC, charge transfer and lithium diffusion inside the respective electrodes can also be studied in these time domains.^{1–4} However, since some data points at low (< 1 Hz) or high (> 30 kHz) frequencies exhibited systematic errors in Kramers-Kronig transformation (please also see Answer#32) the evaluated frequency range was confined to 30 kHz – 1 Hz.

The fact that the literature gives such a range, and all others measure it is not a scientific argument. Furthermore, the choice of the range of frequencies is strongly related to the type of chemistry of the battery and, most importantly, the electrochemical processes which occur.

...For the low frequency impedance, a slight drift is observed over the course of two hours.

I wish the authors had included comments on observed drift and the frequencies that could be represented in the complex plane plots to assist readers who may be less experienced with EIS analysis, particularly given that the reliable measurement range is limited to 30 kHz to 1 Hz.

Figures S20, S25, S27: Equivalent circuit fit: The use of the Rs-CPE model in series appears unwarranted. The Kramers-Kronig analysis reveals that the range between 30 kHz and 1 Hz spans only four decades, resulting in a limited number of data points. This scarcity makes it difficult to draw meaningful conclusions or achieve reliable fitting. Furthermore, the absence of data at both lower and higher frequencies complicates the identification and validation of the CPE's presence. Figures S20, S25 and S27 should be removed.

Version 2:

Reviewer comments:

Reviewer #1

(Remarks to the Author)

Overall, the manuscript has improved since the last iteration. There is still some lack of clarity surrounding the exact nature of the alloy phase formed, due to the reliance on NMR. Likewise, some simple ^{19}F NMR would also show that the SrF₂-PVDF-HFP is not consuming Li.

I also disagree with the rebuttal statement:

'Echo-based pulse sequences can act as T2 filters for fast relaxing species (e.g. lithium metal), which limits the quantitative interpretation of NMR spectra.'

Spin-echo NMR spectroscopy is routinely used to record wideline paramagnetic, magnetic, and metallic NMR signals. Counter to the statement above, the lack of echo could mean broader resonances are being filtered during acquisition.

Reviewer #2

(Remarks to the Author)

Review Report Manuscript 513991_2_art_file_10492359_st6fzz

The authors did further improve their manuscript "Origins of lithium inventory reversibility with alloying functional layers in 'anode-free' lithium metal batteries". Especially the addition of ex-situ and operando NMR experiments to corroborate the formation of an Li_xSn alloying layer on the Cu current collector improved the quality of the manuscript significantly. I recommend publishing after addressing the following minor remarks:

1. Previous Fig. S10 was dismissed and as it seems not replaced or replaced by Figure S27? The numbering in the manuscript and in the SI are not consistently changed. Please check carefully.
2. Page 19 states that MAS NMR of scraped powder from negative electrodes was not feasible. Instead, NMR analyses of complete negative electrodes were conducted. Please clarify that these were done in a wideline setup (no MAS). This at the moment remains unclear in the manuscript.

Reviewer #3

(Remarks to the Author)

Dear Editor and Reviewers,

we would like to express our gratitude for the careful examination of our manuscript and inviting a major revision. The comments and remarks from the reviewers were helpful to spot the sections of our manuscript that required more explanation and additional data, improving the overall quality of our study. We hope that we were able to address all concerns by revising and rephrasing parts of the manuscript and also carrying out additional experiments.

Following the valued suggestions of the reviewers, we prepared a point to point response and have highlighted changes in the manuscript and supporting information in yellow.

Reviewer #1 (Remarks to the Author):

The manuscript by Wichmann and colleagues outlines an NMR method to quantify capacity loss in anode-less cells. The method is then utilized to determine the mechanism of lithium reversibility on bare Li and coated current collectors. The script presents interesting methodology and observations, however, it suffers from the limited SEI/interphase structural data to corroborate the findings. For example, the authors postulate that lithium metal embedded in the matrix benefits from reduced SEI layers, some evidence of this from another technique is required before any firm conclusion can be drawn. Currently, this paper is limited by solely presenting an observation.

Comment #1: The authors link enhanced interfacial transport rather than improved morphology of the lithium metal deposits. But there is no evidence to support that the morphology is not a factor in reduced isolated lithium deposits. The paper requires some evidence that the morphology is consistent in both the coated and bare cells.

Answer #1: The assessment of lithium deposit morphology by ^7Li NMR peak deconvolution is a well-established method in the literature and may even be considered superior to available optical methods such as SEM, since it captures the full electrode and is non-destructive. While the morphology evaluation for lithium deposits via deconvolution of ^7Li NMR spectra has been validated for bare copper in NMR pouch cells by utilizing SEM in the initial version of our manuscript, we now added top-view (**Figure 4 c,d** and **Figure S17**) as well as cross-section SEM data (**Figure S14, S15, S16, S29, S30** and **S31**) of lithium deposition on the coated copper in coin cells to support our claims on lithium morphology in the revised manuscript and supporting information.

Comment #2: In the coated cells, the script should outline where the is Li plated?

Answer #2: Visualization of lithium deposits on the coated copper using cross section SEM has been added to the revised supporting information (also see Answer#1).

Comment #3: The script states no ^7Li signal corresponding to Sn/Sr alloys is observed. The Sn alloy would appear between 0-100 ppm (J. Phys. Chem. C 2010, 114, 14, 6749–6754). This region of the spectra should be presented in the SI to support this statement.

Answer #3: Additional ^7Li NMR spectra with acquisition parameters that are optimized for the detection of Sn/Sr-Li alloys have been recorded after charge, discharge and a constant voltage discharge step, respectively, for NMR pouch cells containing bare and coated copper (**Figure S10**). Here, a broad peak in a frequency region that was previously reported for highly lithiated Li-Sn alloys is not only observed with coated but also and even more intense with bare copper. Thus, we ascribe these to lithium ions in interphases or diffused into the copper current collector rather than the alloys. Most likely, the amount of such potentially formed alloys (reflecting 0.05 – 0.06 mAh/cm²) is below the limit of detection.

Comment #4: The ^{19}F NMR could also be used to show that the coating is not affected by the insertion of Li.

Answer #4: We agree, but unfortunately, the coil of our custom-made NMR probe for analysis of pouch cells is set to the ^7Li frequency (77.78 MHz) and cannot be tuned to the frequency of ^{19}F (188.30 MHz) at our magnetic field strength of 4.7 T.

Comment #5: Some of the figures contain more data than the accompanying key. For example, Fig 4c the key has 6 data points and the figure contains far more than these 6. The same issue is seen in Figure 5. It is difficult to review the images until the reason for these additional data points is given.

Answer #5: To display the reproducibility of our experiments we decided to always plot all data acquired for nominally identical experiments. Thus, every key entry has at least two sets of data in most plots. A clarification has been added to the figure descriptions.

Comment #6: What are the effects of the Cu current collector on the NMR signal attenuation?

Answer #6: The signal attenuation for lithium metal is just minorly prolonged by the Cu current collector. A comparison of FIDs between a lithium stripe and lithium on copper from the same supplier has been added to the revised supporting information (**Figure S5 a**).

Comment #7: Quantitative NMR requires knowledge of the recycle delay, this is not presented.

Answer #7: Proof of calibration for the recycle delay as well as a pulse optimization for the detection of lithium metal on copper via ^7Li NMR was added to the supporting information (**Figure S4 a and b**).

Comment #8: The NMR figures are presented as intensity against shift, this presumes the FWHM remains constant. The integrated area would be more appropriate.

Answer #8: Thank you for the comment, the evaluation of the ^7Li NMR data was already done by integrated area. We agree that the axis labels were misleading, so all the figures containing integrated NMR data were adjusted accordingly.

Comment #9: The authors use the terms upfield/downfield, this terminology is only relevant if the authors used a field-swept instrument. These should be changed to lower/higher frequency or in terms of the Knight shift.

Answer #9: We agree that the terminology of up/downfield shift was misleading and rephrased it in all cases.

Comment #10: How is the reduction in Li metal intensity due to cell disassembly controlled? Can the authors take the NMR of the separator to show the unaccounted-for Li?

Answer #10: We conducted new experiments, in which NMR pouch cells for the calibration measurements were disassembled and solely the separators were measured. Even when measuring three separators from disassembled cells simultaneously, no lithium metal signal is observed, as shown in **Figure S5b**.

Comment #11: The errors in the quantitation by NMR, how are they determined?

Answer #11: Each NMR measurement with bare and coated copper in a respective cycle has at least been reproduced once. The obtained data sets for nominally identical experiments were evaluated separately, yielding their average as the value for the data point and the standard deviation between measurements as the error margin. This was also clarified in the experimental section.

Comment #12: The authors should fit the Nyquist plots in the SI.

Answer #12: An equivalent circuit fit for the most relevant Nyquist plots as well as the obtained resistances was added to the supporting information (**Figure S20, S25 and S27** as well as **Table S1, S2 and S3**). Since in some cases, the semi-circle is not fully resolved in the applied frequency range, the actual values for interfacial resistances should only be evaluated with caution. Therefore, we decided to discuss the trends observed in impedance data qualitatively instead of relying on explicit values derived from equivalent circuit fits. Nevertheless, the good agreement of the experimental data and equivalent circuit fits support our interpretation of two processes being displayed in the employed frequency range.

Comment #13: There are statements within the text that require referencing, such as line 105-107.

Answer #13: Further references were added to the manuscript.

Comment #14: Many of the reference sources are 'not found', therefore it is difficult to comment on whether the script is referenced appropriately.

Answer #14: We apologize for an error that occurred upon creating the pdf file for submission. The reference sources are now properly assigned in the revised manuscript.

Reviewer #2 (Remarks to the Author):

The authors present a detailed in-situ ⁷Li NMR spectroscopic study combined with impedance spectroscopy of an “anode-free” Li metal battery to unravel the distribution of Li in the different battery components, such as cathode, in-situ formed anode, interlayers and the formation of isolated Li deposits, as a function of SOC and SOH. “Anode-free” Li metal batteries are a very promising battery design for future applications, combining the possibility to boost energy density and at the same time limiting safety hazards and costs by reducing the Li metal content to the volume/mass stored in the positive electrode. As such, the presented study is of high relevance for a better understanding of the mechanisms during Li plating and stripping at the negative current collector and thus for the further development of “anode-free” Li metal batteries.

The authors present comprehensive and high-quality in-situ ⁷Li NMR data on six “anode-free” Li metal battery systems, which used NMC, LMNO or LFP as cathode materials against Cu or coated Cu with an alloying functional layer as negative current collectors. The authors provide a careful analysis of ⁷Li NMR spectra to quantify the Li distribution in different compartments as a function of (dis)charge cycles. Furthermore, the authors investigate the morphology of Li deposits using ⁷Li NMR and SEM. Finally, impedance spectroscopy is used to characterize transport properties and derive a mechanism for dead Li in LHCE.

The authors carefully evaluated the influence of paramagnetism of NMC cathode material as a function of SOC on both ⁷Li chemical shifts as well as signal intensity for quantitative analyses. In this way the authors are able to quantify the amount of dead lithium (electronically isolated Li metal), and the formation of interlayers (CEI and SEI) during cycling of the batteries. Concerning this procedure, I would like to ask the authors to clarify a few minor things:

Comment #15: In Fig. 1 signal intensity was used as y axes label, which could either refer to relative height of the signals or their integral (which I guess was meant). Please clarify.

Answer #15: Evaluation of ^7Li NMR data was already done by integrated area but we agree that the axis labels were misleading. All figures were adjusted accordingly.

Comment #16: Furthermore, in Fig.1 the difference between light and darker color (e.g. light and dark red, light and dark green) spectra is not explained. Please also clarify this.

Answer #16: To display the reproducibility of our experiments we decided to always plot all data acquired for nominally identical experiments. Thus, every key entry has at least two sets of data in most plots. A clarification has been added to the figure descriptions.

The authors then show that ^7Li NMR in combination with absolute capacity losses can be used to quantify the amounts of kinetically hindered and dead lithium compared to active “strippable” Li as a function of (dis)charge cycles.

Comment #17: In Figure 2 a and b) significantly higher chemical shift were observed for Li deposits in both charged and discharged states (between 250 and 280 ppm) compared to the shifts shown in Fig.1 (between 240 and 250 ppm). This currently remains unclear in the manuscript.

Answer #17: We thank the reviewer for noticing this issue. The visualization of our method displayed in Figure 2 was plotted using data acquired with a carbonate-based electrolyte instead of the LHCE used in most of the experiments for our manuscript. We agree that this is misleading and used data with an LHCE to display our methodology.

The authors proceed with comparing Li plating on Cu vs. a coated Cu with $\text{SrF}_2+\text{PVDF HFP}@Sn@Cu$, with the latter clearly improving the reversibility of Li plating/stripping.

Comment #18: I wish the authors would have included more information on the coating itself. At the current stage there is very little on the state-of-art of this coating or why it was chosen.

Answer #18: A brief review of previous approaches using our or analogous multi-functional alloying coatings on copper has been added to the manuscript.

Comment #19: Page 12, lines 350 to 360: I propose that the authors refer to the experimental details to clarify how the relative amount of “strippable” Li, dead Li and Li in interphases are calculated and converted to capacity losses. Currently, this is hard to follow.

Answer #19: We rephrased this section of our manuscript to clarify our combined evaluation of electrochemical and spectroscopic data.

The authors then show that Cu and coated Cu result in similar Li morphologies starting from dense Li towards mossy and dendritic Li for deposited Li as a function of battery cycling by established ^7Li chemical shifts for the different morphologies and SEM Images.

Comment #20: It could be helpful to indicate dense, mossy and dendritic Li in the SEM Images in Fig. 4a)

Answer #20: The regions with different lithium deposit morphologies have been indicated in the SEM images (Figure 4 b and d).

Comment #21: Is the morphology the reason for the shift difference in Fig.1 and Fig.2 (first dense and later mossy/dendritic Li-compare comment 3.)? This should then be stated somewhere in Fig.2 to avoid confusion.

Answer #21: No, see Answer #17.

And finally, the authors state that no Li signal was observed in either Sn or Sr alloys.

Comment #22: At least for Li_xSn a very different shift range (between 0 and 100 ppm) is expected (compare J. Phys. Chem. C 2010, 114, 6749–6754). Did the authors vary the offset to test for these shifts as only very low B1 field strength was used in the study? If the authors did, these spectra should be included in the SI.

Answer #22: Novel ^7Li NMR spectra with acquisition parameters optimized for the detection of Sn/Sr-Li alloys have been recorded after charge, discharge and the CV step for NMR pouch cells containing bare and coated copper (Figure S10). Here, a broad peak in a frequency region that was previously reported for highly lithiated Li-Sn alloys is not only observed with coated but also with bare copper. Thus, we ascribe these to lithium ions present in interphases or that diffused into the copper current collector rather than alloys. Most likely, the amount of such alloys (reflecting 0.05 – 0.06 mAh/cm²) is below the limit of detection.

Comment #23: How thin is the coating? Would it be possible to record ^{119}Sn NMR spectra of the coating itself? Further insight into how the coating itself changes as a function of cycling might help to explain the role of the coating for the improved Li inventory reversibility and might provide directions for future coatings. It could further strengthen the postulation “We postulate that lithium metal embedded within the alloying polymer matrix benefits from less accumulation of SEI layers on the outside of lithium deposits and an improved interconnection by the alloying metals within the metal deposits, which then translates into comparatively low interfacial resistances in the discharged state and a reduced formation of isolated ‘dead’ lithium deposits (Figure 7).” (page 21) and support the findings from impedance spectroscopy.

Answer #23: Further characterization of the coating in initial and later stages of cycling have been added to the manuscript (see Answer #22). Since the gyromagnetic ratio and natural abundance of ^{119}Sn are worse than for ^7Li and no alloys were observed with ^7Li , ^{119}Sn experiments were not carried out.

Furthermore, I have a few more detailed, mostly minor suggestions/comments:

Comment#24: Please make sure that all abbreviations, such as LFP, LMNO, SOC, etc. are defined upon their first occurrence. This will make it easier for a broad readership to follow the manuscript.

Answer #24: All abbreviations are now defined upon their first occurrence.

Comment #25: Page 14, lines line 411, please provide a reference for “previously defined chemical shift boundaries”

Answer #25: References for the definition of chemical shift boundaries for different lithium deposit morphologies were added.

Comment #26: Page 14, line 413: was the FWHM of the three Voigt functions fixed? If yes, on which bases?

Answer #26: We thank the reviewer for this suggestion. The width of the three Voigt functions representing different lithium deposit morphologies was constrained to not exceed the width obtained for dense, commercial lithium metal electrodes in vicinity of NMC622 electrodes at 100% SOC in the revised manuscript. The respective sections in the experimental as well as main section of the manuscript were rephrased .

Comment #27: Page 16, line 440: please define the abbreviation HSAL.

Answer #27: High-surface-area lithium has been defined in the revised manuscript.

Comment #28: Page 21, line 583: “alloying polymer matrix” seems misleading as Sn or Sr are alloying with Li and not the polymer, which is only the binder. Correct? (more details on the coating would be useful – compare comment 4)

Answer #28: We thank the reviewer for noticing this misleading formulation and clarified it in the revised manuscript.

Reviewer #3 (Remarks to the Author):

Manuscript Number: NCOMMS-24-33721

The manuscript "Origins of lithium inventory reversibility with alloying functional layers and a localized high concentrated electrolyte in 'anode-free' lithium metal batteries" by Wichmann et al. explores the reasons behind improved metal inventory reversibility in 'anode-free' lithium metal batteries with a multi-functional alloying layer on the copper current collector. The study highlights two key findings: Quantitative ^7Li NMR spectroscopy shows that accelerated formation of a stabilized Solid Electrolyte Interphase (SEI) and reduced electronically insulated Li deposits are crucial for enhanced 'anode-free' lithium metal battery longevity. Additionally, the reduced formation of isolated lithium deposits is attributed to improved interfacial transport properties rather than better lithium deposit morphology. Wichmann et al. primarily used two techniques in the study: static NMR spectroscopy and electrochemical impedance spectroscopy (EIS). The authors utilized a two-electrode setup in standard coin-cell and pouch-cell geometries and for the three-electrode setups they employed a PAT-Cells configuration with a coated lithium ring as the reference electrode, as well as a custom three-electrode T-cell (Swagelok) where both the counter and reference electrodes are lithium, and the working electrode is commercial LiO on Cu. The study compared two-electrode setup systems using coated (multi-functional alloying layer: Sn, SrF₂ and PVDF-HFP) and uncoated copper sheets.

The conclusions throughout the manuscript are predominantly based on the NMC622 composite electrode chemistry using localized high-concentration electrolytes (LHCE) in 'anode-free' lithium metal battery systems.

Comment #29: In Figure 2b), NMR-based evaluation lacks evidence for other chemistry-related measurements. This raises the question of why excess lithium was not detected in other systems, such as LMNO or LPF composite electrodes (there are references to this in the literature), to which it could potentially be linked? Additionally, the effect of different electrolytes is poorly addressed. Even though the authors mention that each individual cell setup requires separate calibration, there is a lack of supporting evidence and reference experiments performed.

Answer #29: A different electrolyte was used for these experiments in order to allow cycling up to 4.95 V in the case of LNMO positive electrodes. We agree that the difference in chemical shifts and

capacity losses obtained with this electrolyte compared to LHCE might confuse the reader. Thus, we moved the comparison of NMR data evaluation for different positive electrodes to the SI.

To accurately quantify capacity losses with LFP and LNMO, separate calibrations would indeed be necessary. However, the comparison between the positive electrodes is intended as an illustrative example, showing that with NMC positive electrodes a differentiation between dead and excess lithium is necessary. Only with this additional step comparable dead lithium intensities for all positive electrodes are observed. Due to the differences in positive electrode para-magnetism between LFP, LNMO and NMC, a straightforward evaluation of integrals might indeed be incorrect. Thus, instead of a capacity evaluation based on the integrals, which was depicted in the initial version of the manuscript, we only plot the spectra to display the peculiarities with NMC positive electrodes.

Kinetically limited capacity is not detected with LFP and LNMO as there is no such capacity present in 'anode-free' cell setup using these positive electrodes. In conventional lithium metal batteries, where 100% re-lithiation is possible, due to excessive lithium from the negative electrode, these electrodes display only very little kinetically limited capacity (**Figure S8**). In 'anode-free' cell systems, kinetically limited capacity may remain at the negative electrode creating an excess lithium reservoir. However, the irreversible capacities from the negative electrode, which are not observed as irreversible capacity in conventional lithium metal batteries due to an excess of lithium metal, already consume the kinetically limited "excess" capacity from the positive electrode. Only when the kinetically limited capacity is larger than the initial capacity losses at the negative electrode, an excess lithium reservoir will remain. This is certainly the case with NMC positive electrodes, where the initial Coulombic efficiencies for conventional and 'anode-free' LMBs are almost identical and limited by the positive electrode in both cases. With LFP and LNMO, higher initial Coulombic efficiencies are possible with an excess lithium metal reservoir (see conventional lithium metal batteries), but these are not achieved in 'anode-free' lithium metal batteries, which indicates that the little kinetically limited capacity is instantly consumed by parasitic reactions. We also clarified these differences between the positive electrodes in the revised manuscript.

Comment #30: The manuscript neglects fundamental pillars that influence the data interpretation, such as the test cell orientation-effect bulk magnetic susceptibility (BMS) shifts (Figures 1,2), in addition to the 7Li signals. It is well established that the shape of the sample, the packing of the material, and the sample's orientation relative to B_0 contribute to the variable effects of BMS. Thus, due to the different geometries used, results from a two-electrode system cannot be directly compared/transferred to those from a three-electrode system and vice versa. The same is valid for EIS analysis.

Answer #30: The introduction has been revised to include effects from cell orientation and bulk magnetic susceptibility effects. Since for NMR experiments solely a two-electrode setup with identical pouch cell geometry was employed and all cells had a mechanically fixed orientation with respect to the magnetic field, no issues for comparability arise for interpretation of the NMR data.

For EIS analysis, different cell geometries and setups are compared. Here, the differences are pointed out but the impedance features of interest are similar. To avoid confusion for the reader, assignment of data to the different cell setups has been clarified throughout the manuscript. Furthermore, an overview of cell setups and their geometry has been added to the supporting information (**Figure S1** and **S2**).

Comment #31: It should also be noted that electrochemical tests have been planned in the two-electrode and three-electrode configurations, which is an excellent start. However, the interpretation of conducted EIS experiments is questionable. The authors do not master this domain and the EIS proves the lack of understanding of this experimental method and interpretation of its

results. EIS complex impedance plot must be squared and ortho-normed. It remains unclear why the authors use the electrochemical impedance spectroscopy method in this study. What processes were they hoping to detect in the frequency range 300 kHz - 0.1 Hz?

Answer #31: Nyquist plots depicted in the manuscript are ortho-normed. By ensuring identical scaling of the x and y axes, they are also squared. To avoid parts of the plot not containing data, x and y axes were only shown to the maximum value of the respective Z' and $-Z''$ values. To display the validity of our data representation, a completely squared graph for impedance data reported in our manuscript is compared to our way of impedance data presentation.

Furthermore, the frequency range between 300 kHz and 0.1 Hz is commonly employed in battery related research. Typically, electrolyte resistance, contact resistances, grain boundary as well as interfacial processes are observable. Depending on the cell chemistry and SOC, charge transfer and lithium diffusion inside the respective electrodes can also be studied in these time domains.¹⁻⁴ However, since some data points at low (< 1 Hz) or high (> 30 kHz) frequencies exhibited systematic errors in Kramers-Kronig transformation (please also see Answer#32) the evaluated frequency range was confined to 30 kHz – 1 Hz.

Comment #32: Experiments conducted are not performed under steady-state conditions and do not follow the rules of linearity, stability, causal and finite. follow the rules of linearity, stability, causal and finite. A. Lasia / Electrochemical Impedance Spectroscopy and its Applications, doi; 10.1007/978-1-4614-8933-7

Answer #32: We would like to thank the reviewer for suggesting literature to check and proof the validity of the obtained EIS data. We conducted additional experiments and Kramers-Kronig transformations on our NMR Pouch cells to address these requirements, the results are displayed in the supporting information (**Figure S18, S19, S22 and S26**).

Kramers-Kronig transformations revealed systematic instead of random noise^{5,6} differences occurring for the EIS data at frequencies < 1 Hz and > 30 kHz, especially data obtained in three-electrode and dynamic EIS measurements. Furthermore, impedance spectra recorded after the constant voltage discharge step displayed severe differences between experimental and transformed data. Thus, the impedance data obtained after the CV step were discarded for the discussion. Other data sets were confined to the frequency range of 30 kHz – 1 Hz for the evaluation. Here, only random errors with low magnitude (< 4 % relative residuals in all cases) were observed. Since the transformed data agrees with the experimental data, EIS data is formally correct and indeed Kramers-Kronig compliant, meaning that the requirements for linearity, stability, causality and finiteness are met. Nevertheless, we performed additional experiments concerning stability and linearity.

Stability: EIS experiments were carried out repetitively over a time frame of 2 h. In the charged and discharged state, the spectra are almost identical in the high frequency region. For the low frequency impedance, a slight drift is observed over the course of two hours. However, since lithium metal is a highly reactive electrode, reactions between the electrolyte and electrode are expected to occur continuously, especially with a liquid electrolyte. Thus, obtaining completely stable impedance in these systems is very challenging. Since only minor drift is observed after 2 h, we consider our EIS data reasonably stable for our qualitative interpretation of data.

Linearity: Decreasing the perturbation amplitude twice still yields identical impedance data in the charged state. In the discharged state, the EIS spectrum as a whole remains similar but exhibits a minor but tolerable decrease in low frequency impedance. Since electrochemical systems are usually highly non-linear and only linearized with a small amplitude and some of the low frequency drift observed is also due to the limited stability, we consider the investigated systems to be sufficiently linear.

Since our use of EIS in the manuscript is not designed to deduce actual values for the interfacial or charge transfer resistances but instead to discuss global trends for EIS spectra, depending on the negative electrode and the batteries state of charge, the criteria of linearity and stability in our view are sufficiently met.

In dynamic impedance measurements, constraints of linearity and stability cannot be met, since the cell is under operation and not in a stationary state. Nevertheless, by choosing an appropriate charging rate for the excitation time scale, quasi-stationary system conditions are achieved.^{7,8} Previous studies have also reported sufficient stability for DEIS with a lower frequency limit of 0.1 Hz and a discharge rate even higher than in our case (2A/0.4C compared to 0.05 mA/cm²/0.025C).⁹ The combination of lower frequency limit and DC current applied for our DEIS study are well within the boundaries defined and used in previous literature.^{8,10}

Comment #33: Furthermore, in a two-electrode system, it is very misleading to assume that impedance spectroscopy only separates information for the negative or positive electrode. Even if it is measured in a totally charged or discharged state, and one might just as well assume that the total impedance of a cell is now determined by only one electrode, this is not the case. Batteries are not electric circuits, so the impedance at the complex level of the cell is determined by several factors.

Answer #33: We are aware that the EIS data of two electrode cells always contains information about processes at both electrodes. By applying a reference electrode, the contributions from positive and negative electrode can be separated. We rephrased our previous interpretation of EIS data to clarify that one of the electrodes dominates the spectrum in a two-electrode setup rather than solely determining it. However, for the utilized three electrode setup, the separation of positive and negative electrode impedance is suitable, as already shown in literature.¹¹

Comment #34: The position of the reference electrode concerning the working electrode is of great importance. Unfortunately, it is not discussed in the manuscript. Without validating different cell geometries and implementing the reference electrode (RE), including factors related to the chosen material, position, and design of the RE, artefacts can be introduced, as several factors can influence the measured data. Additionally, demonstrating the quality of EIS measurements using Kramers-Kronig relations is necessary.

Answer #34: The results obtained from EIS with different cell geometries are compared to each other, with the impedance feature of interest being similar between the utilized cells. Also, validity of PAT cells for three electrode impedance spectroscopy measurements has been shown.¹¹ With this cell geometry, particularly the electrode alignment and position of the reference electrode, a

symmetrical field distribution is ensured. Since we did not design or develop the cell format utilized for three-electrode cells and it was validated already in the literature¹¹, we did not want to depict the valid separation of electrode responses to the impedance perturbation as an achievement accomplished by our research group. Therefore, we solely added a description of the cell geometry and components used for three-electrode cycling and EIS measurements (**Figure S2**) but refrained from a detailed discussion. For Kramers-Kronig relations, please see Answer#33.

Comment #35: “Consistent with the impedance data from NMR pouch cells in a two-electrode configuration, similar DRT spectra are observed in the initial cycles with both negative electrodes.”

Please note, that the DRT method is generally unsuitable for systems with blocking electrodes, such as those encountered in batteries. Even if DRT analysis was used, it is necessary to fully assess the accuracy and resolution of the DRT inversion obtained from the time measurements. In summary, the EIS experiments and analysis do not support the claim that “the reduced formation of isolated lithium deposits is attributed to improved interfacial transport properties rather than better lithium deposit morphology.”

Answer #35: After further studying literature on DRT, we agree that DRT is not suitable for blocking electrodes. However, electrodes used in batteries are considered blocking electrodes only in certain conditions. For NMC, this is the case at full lithiation after a constant voltage step, since no charge transfer may take place. With the lithium reservoir being limited to the positive electrode for ‘anode-free’ lithium metal batteries, complete re-lithiation is not possible considering the occurrence of capacity losses. A negative electrode in ‘anode-free’ lithium metal batteries may only be considered a blocking electrode if all electrochemically active lithium has been removed, impeding charge transfer.

Though these conditions do not apply for most of our impedance data, we still decided to refrain from using DRT analysis for interpretation of impedance data in the revised version of the manuscript. Since DRT analysis is highly sensitive to the frequency limits, some of the trends observable in Nyquist plots are not represented in DRT analysis. Furthermore, we wanted to not overcomplicate the discussion of impedance data by comparing different forms of data representation to each other. We hope the consistency improves the reading flow of our manuscript.

Comment #36: What was the contact pressure used for assembling the test cell with various thicknesses of the NMC622, LMNO, LPF vs coated (multi-functional alloying layer: Sn, SrF₂ and PVDF-HFP) and uncoated copper sheets?

Answer #36: We assume the contact pressure to be relatively comparable between the utilized cell setups. Especially for NMC622 || Cu and NMC622 || Coated Cu, which most of our work is focused on, and which are the only cell setups where electrochemical performances are compared, the differences in thickness is only about 2-4 μm , which we believe to result in marginally differences for contact pressure. While the difference in electrode thickness between the mentioned positive electrodes is on the order of tens of μm , we assume the contact pressure to be comparable here as well, but we also merely compare results for these cells on a qualitative basis.

Comment #37: What was the final thickness of the electrode in the re-assembled test-cell? These physical (thickness) properties clearly do not manifest as noticeable differences in electrochemical behaviour when measured in a three-electrode setup, why?

Answer #37: We are not certain, what the reviewer refers to with this comment. Cells were solely re-assembled for calibrating NMR measurements for the impact of NMC electrodes in NMR pouch cells,

which are a two-electrode setup. In three-electrode cells, differences in thickness between copper and coated copper are in the range of 2-4 μm . We would not expect these differences in thickness to affect the electrochemical behavior.

Comment #38: The data are incomplete, and the execution and analysis leave much to be desired. In summary, the electrochemical measurements are not convincing, and the data analysis does not meet the relevant criteria of Nature Communications. This manuscript still needs careful editing by writers with expertise in electrochemistry.

Answer #38: Upon revision, we have augmented the provided electrochemical data and substantiated both the analysis and interpretation thereof, thereby clearly corroborating the reliability of our work.

References

- (1) Raccichini, R.; Amores, M.; Hinds, G. Critical Review of the Use of Reference Electrodes in Li-Ion Batteries: A Diagnostic Perspective. *Batteries* **2019**, *5* (1), 12. DOI: 10.3390/batteries5010012.
- (2) Gaberscek, M.; Moskon, J.; Erjavec, B.; Dominko, R.; Jamnik, J. The Importance of Interphase Contacts in Li Ion Electrodes: The Meaning of the High-Frequency Impedance Arc. *Electrochem. Solid-State Lett.* **2008**, *11* (10), A170. DOI: 10.1149/1.2964220.
- (3) Atebamba, J.-M.; Moskon, J.; Pejovnik, S.; Gaberscek, M. On the Interpretation of Measured Impedance Spectra of Insertion Cathodes for Lithium-Ion Batteries. *J. Electrochem. Soc.* **2010**, *157* (11), A1218. DOI: 10.1149/1.3489353.
- (4) Abarbanel, D. W.; Nelson, K. J.; Dahn, J. R. Exploring Impedance Growth in High Voltage NMC/Graphite Li-Ion Cells Using a Transmission Line Model. *J. Electrochem. Soc.* **2016**, *163* (3), A522-A529. DOI: 10.1149/2.0901603jes.
- (5) Boukamp, B. A. A Linear Kronig-Kramers Transform Test for Immittance Data Validation. *J. Electrochem. Soc.* **1995**, *142* (6), 1885–1894. DOI: 10.1149/1.2044210.
- (6) Lasia, A. *Electrochemical Impedance Spectroscopy and its Applications*; Springer New York, 2014. DOI: 10.1007/978-1-4614-8933-7.
- (7) Meddings, N.; Heinrich, M.; Overney, F.; Lee, J.-S.; Ruiz, V.; Napolitano, E.; Seitz, S.; Hinds, G.; Raccichini, R.; Gaberšček, M.; Park, J. Application of electrochemical impedance spectroscopy to commercial Li-ion cells: A review. *Journal of Power Sources* **2020**, *480*, 228742. DOI: 10.1016/j.jpowsour.2020.228742.
- (8) Koseoglou, M.; Tsioumas, E.; Ferentinou, D.; Jabbour, N.; Papagiannis, D.; Mademlis, C. Lithium plating detection using dynamic electrochemical impedance spectroscopy in lithium-ion batteries. *Journal of Power Sources* **2021**, *512*, 230508. DOI: 10.1016/j.jpowsour.2021.230508.
- (9) An, F.; Chen, L.; Huang, J.; Zhang, J.; Li, P. Rate dependence of cell-to-cell variations of lithium-ion cells. *Scientific reports* **2016**, *6* (1), 35051. DOI: 10.1038/srep35051. Published Online: Oct. 11, 2016.
- (10) Huang, J.; Ge, H.; Li, Z.; Zhang, J. Dynamic Electrochemical Impedance Spectroscopy of a Three-Electrode Lithium-Ion Battery during Pulse Charge and Discharge. *Electrochimica Acta* **2015**, *176*, 311–320. DOI: 10.1016/j.electacta.2015.07.017.
- (11) Wünsch, M.; Füllner, R.; Sauer, D. U. Metrological examination of an impedance model for a porous electrode in cyclic aging using a 3-electrode lithium-ion cell with NMC111 | Graphite. *Journal of Energy Storage* **2018**, *20*, 196–203. DOI: 10.1016/j.est.2018.09.010.

Dear Editor and Reviewers,

We are pleased to see that the first revision of our manuscript resonated well and addressed the majority of concerns. Once again, the constructive feedback is highly appreciated and particularly helpful to identify sections that require rephrasing or further experimental evidence, such as the alloy formation with the coated negative electrodes. We hope that the changes marked in the manuscript successfully amplify its relevance for the scientific community and address the remaining issues.

REVIEWER COMMENTS

Reviewer #1 (Remarks to the Author):

The authors have made substantial changes to the manuscript. However, I still find the presented work convoluted and lacking general applicability. The universal applicability of this methodology is very low due to the substantial amount of experimental corrections (orientation, shift, intensity, FWHM, B1) required to achieve results.

Comment #1:

The discharge capacities achieved here is lower than what other anode-free architectures present in the literature (Nature Energy, 4, 683–689, 2019). If practical capacities >200 mAh/g is employed how much Li is deposited and would this cause a skin depth issue? Without achieving commercially required discharge capacities (or even improvements on current state of the art NMC systems) then I would question the practicality of this system.

Answer #1:

The cited publication does not give a specific capacity in mAh per gram active material, rather an “absolute” capacity of multilayered pouch cells in mAh is stated, without taking the cathode mass into account. Other publications employing NMC622 with similar cathode active mass loading and charging rates also report specific capacities of ≈ 160 mAh/g (Adv. Energy Mat., 14, 5, 2302261, 2024).

When using NMC positive electrodes with increased Ni-content, *e.g.* NMC811, specific capacities of 200 mAh/g can be achieved. However, for the skin-depth effect, the absolute lithium deposition capacity and thus areal capacity and actual weight of the positive electrode are more relevant than its (specific) capacity per weight. We utilized positive electrodes with a capacity of 2.3 mAh/cm², since this is close to commercially relevant areal capacities but was expected to still allow for a quantitative detection of lithium metal with 5 μ m lithium thickness per mAh and the theoretical skin depth limitation of 14.7 μ m at our magnetic field strength. For future works, investigation of multi-layered pouch cells would be even more application-oriented, but requires understanding of single layer pouch cells first, which is presented in this manuscript.

Comment #2:

There is a lack of structural analysis of the SEI leading to broad claims to be made in the manuscript *i.e.* 'The irreversibly formed Sr-Li and Sn-Li alloys' - the authors present no evidence of these alloys. Therefore the mechanism is purely speculative.

Answer#2:

We thank Reviewer#1 for the comment and agree that besides the potential profile of the negative electrode, further (NMR) spectroscopic evidence for the formation of alloys should be provided. Since ^7Li NMR spectroscopy on regular NMR pouch cells utilized throughout the manuscript indicates the presence of Li alloys (see Answer #8) but yields a broad and low intensity signal (Figure S27), we in addition carried out *ex situ* NMR experiments with increased amounts of compounds and at higher magnetic field to enhance the recorded signal intensity (see Figure 6 and Figure S28). Also, *in operando* and *in situ* experiments (Figure 7) were conducted to characterize the alloys upon battery operation. We hope that the identification of alloying phases in the functional layer satisfies your desired evidence of alloys.

Comment #3:

The abstract is rather vague and fails to summarize the findings.

Answer #3:

We thank the reviewer for the remark and rewrote the abstract. We hope that the newly phrased abstract underlines the generally applicable conclusions beyond the specialized NMR methodology.

Comment #4:

The authors should measure T1 rather than the recycle delay.

Answer #4:

The T1 time of metallic Li species is well known and on the order of some seconds, the data in S4 serves as estimate for the delay at which the spectra can be recorded without compromising the peak area to distinguish the metallic Li species.

Comment #5:

Figure S5b is severely truncated and should be rerun with a longer acquisition time. Likewise, S10a should be phased appropriately.

Answer #5:

Since the presented NMR protocol to quantify capacity losses evaluates the NMR signal for metallic lithium species, the parameters to record the FID are optimized for the metallic species, which exhibit faster T_2 -relaxation due to their inherent paramagnetic nature compared to diamagnetic ionic lithium species. While the shorter acquisition time truncates the FID of ionic lithium species, it enables to increase the signal to noise ratio for metallic lithium species in a given measurement time due to an increased number of scans. Though truncation of the FID for ionic species causes sinc(x)-warped lithium ion signals after Fourier transformation (Figure S5b), the baseline in the region of lithium metal is not affected due to the strong knight shift of lithium metal, allowing us to tolerate a diminished ionic lithium signal to enhance metallic lithium signal to noise ratio. Since these parameters were used for all the quantitative NMR measurements, the spectra depicted in Figure S5b were not recorded with a

longer acquisition time in view of continuity. To decrease the impact of FID truncation, line-broadening was applied to Figure S5b.

A 1st order phase correction to the spectra previously depicted in Figure S10 was applied. While this disables quantitative evaluation due to a baseline distortion, it gives a first indication about the presence of Sn alloys (Figure S27). As mentioned in answer #2, this initiated further NMR experiments to characterize the alloy formation.

Comment#6:

'the penetration depth of radiofrequency pulses invoked to excite nuclei of interest is calculated to be 14.7 μm . Thus, even cells containing considerably thin lithium metal electrodes with a thickness of 20 μm cannot be investigated quantitatively.'

Are the authors using a flat surface coil? A cylindrical coil would allow 2 x 14.7 μm penetration depth.

Answer #6:

Our custom NMR-probe head indeed uses a cylindrical coil, which theoretically should allow 14.7 μm penetration depth from each side. However, based on other NMR experiments conducted in our group aimed at developing a quantitative NMR protocol applicable to conventional lithium metal batteries, lithium metal electrodes comprised of 20 μm lithium metal on 10 μm copper foil cannot be detected quantitatively with our experimental setup. Since these results are out of the scope of the current manuscript and will be enclosed in a later publication, we prefer to not go into detail in this regard. We rephrased the section in the current manuscript into a less strong statement.

Comment #7:

In Fig S3 are the authors using a permanent magnet? If so, what is the resolution and have they considered using echoes to reduce inhomogeneities?

Answer #7:

Our NMR200 spectrometer invoked for most of the NMR measurements of pouch-type cells uses a permanent magnet at a field strength of 4.7 T. At a sweep width of 5 MHz and digital FID size of 132k points, the spectral resolution is 76 Hz. With solid lithium metal signals displaying peak widths on the order of 1 kHz, spectral resolution as such is of no concern. Echo-based pulse sequences can act as T_2 filters for fast relaxing species (e.g. lithium metal), which limits the quantitative interpretation of NMR spectra. Therefore, these pulse sequences were not applied.

Reviewer #2 (Remarks to the Author):

The authors did improve their manuscript "Origins of lithium inventory reversibility with alloying functional layers in 'anode-free' lithium metal batteries" and clarified my previous concerns. I recommend publishing after addressing the following comments:

I have a few further remarks:

Comment #8:

1. The spectra presented in Fig. S10 are dephased making a judgment of a (broad signal and small) signal for potential alloying to Li_xSn or Li_xSr difficult. It may also be useful to add a baseline at 0 as guide for the eye.

Answer #8:

We thank the reviewer for this suggestion. To obtain both, lithium-ion and metal peak, in-phase, 1st order phase correction is required. Since this distorts the baseline and disables quantitative evaluation of NMR integrals, we refrained from this in the first revision of our manuscript. However, it indeed reveals a small and broad peak at 95 ppm, which could correspond to the highest lithiated Sn-Li phase, according to previous literature (J. Phys. Chem. C 2010, 114, 14, 6749–6754). To unambiguously identify the respective alloy, we carried out additional NMR experiments with increased amounts of compounds and at higher magnetic field (NMR600) to enhance the signal intensity (see Figure 6 and Figure S28). Also, *operando* and *in situ* experiments (cf. Figure 7) were conducted to characterize the formation of alloys upon battery operation. We hope that the identification of the alloying phases in the functional layer presented here serves as sufficient evidence of the presence of the alloys.

Comment #9:

In the manuscript:

Compared to bare copper, the loss of capacity due to interphase formation is higher initially but the ongoing lithium consumption due to interphase reformation throughout cycling occurs at a lower rate. We interpret this behavior as an accelerated formation of interphases with improved durability.

2. To unambiguously differentiate in SEI formation and durability, one would need the 7Li data for coated Cu after 45 cycles. After 15 and 30 cycles the amount of SEI is similar for Cu and coated Cu within the error ranges. I feel that the statement made is too strong for the current data bases. Additionally, the R2 value of 99.9 % for the fit of the orange squares (interphase formation on Cu) doesn't seem to be correct considering the outlier after 30 cycles. Please correct.

Answer #9:

We thank the reviewer for the suggestion. We consider the trend in SEI capacity as significant enough, since the initial loss is much higher with coated copper ($8.2 \pm 0.6 \%$ vs. $4.8 \pm 0.1 \%$) but levels off afterwards. Indeed, the loss is similar within the error margins after 15 cycles ($14.5 \pm 0.9 \%$ vs. $15.4 \pm 0.8 \%$) but arguably lower after 30 cycles ($18.9 \pm 0.2 \%$ vs. $20.5 \pm 1.0 \%$). We rephrased the respective sections in the manuscript into a less strong statement.

Regarding the R2, the depicted values were determined by linear regression using Origin software when also including the error margins of each data point. We agree that the values should be lower considering the outliers and thus recalculated R2 without the error margin, just using the averaged datapoints. The corresponding plot has been updated.

Comment #10:

In the manuscript:

In contrast to the loss of lithium due to interphase formation, a reduced amount of 'dead' lithium formation when using the coated copper is already observable in the initial cycle.

3. Fig. 3d states 0% and 0% dead lithium formation for both Cu and coated Cu. It would be beneficial to add the exact numbers at least in the text, to make the difference obvious. A direct comparison of the ⁷Li spectra for coated and uncoated Cu (in the SI) after the 1st cycle could also help in this respect. In Fig. S9 I cannot make out any significant difference – I am not sure whether this is due to the representation of the data, or whether there is no significant difference between the spectra. Please clarify.

Answer #10:

Indeed, the scaling in Fig. S9 was not suitable for comparison of the initial cycle. Nominally, the amount of 'dead' lithium is slightly lower with coated copper (<0.1 % versus 0.4 %), as observable in the magnified view of the NMR spectra (Figure S9). We agree that this difference is not significant enough to rely on its interpretation, thus we rephrased the corresponding section. Instead of mentioning differences in 'dead' lithium formation in the initial cycle, the comparison in the manuscript now focuses on cycles 15 and 30, respectively.

Comment #11:

In the manuscript:

We conclude that the cells containing coated copper exhibit accelerated and less resistive interfacial transport upon dissolution of the lithium metal deposits in contact with the current collector. Here, irreversible alloy formation provides the functional layer with facile lithium conduction.

4. An alloy formation is one of the main conclusions from the paper, however direct experimental proof is missing. I agree with the authors that low Li contents in addition to low amounts of coating may cause that the amount of formed alloys is below the NMR detection limit. However, as alloying seems to be the main point of the paper, I feel that direct evidence for the formation of the formed alloys is necessary. One could think about a model system, where the material used for the coating is artificially increased, so that at least the formation of alloys can be proven

Answer #11:

We thank the reviewer for this constructive remark and agree that further characterization of the alloy was required. Since a model electrode with increased thickness of coating might display different electrochemical properties, we enhanced the intensity by increasing the amount of substance and the magnetic field (see Answer #2 and #8).

Reviewer #3 (Remarks to the Author):

The revised manuscript "Origins of lithium inventory reversibility with alloying functional layers and a localized high concentrated electrolyte in 'anode-free' lithium metal batteries" by Wichmann et al. contains changes. Still, unfortunately, the substance of the arguments have not been addressed.

Comment #12:

1) "By combining impedance and ^7Li NMR spectroscopy, the reduced formation of isolated lithium deposits is found to be facilitated by enhanced interfacial transport properties rather than a better morphology of metallic lithium deposits."

This claim is still not covered, nor can it be proven by the tests performed by the EIS, so the word impedance should be removed from this sentence in the manuscript.

Answer #12:

We thank the reviewer for noticing the misleading formulation and rephrased the sentence.

Comment #13:

2) ...What processes were they hoping to detect in the frequency range 300 kHz - 0.1 Hz?

...Furthermore, the frequency range between 300 kHz and 0.1 Hz is commonly employed in battery related research. Typically, electrolyte resistance, contact resistances, grain boundary as well as interfacial processes are observable. Depending on the cell chemistry and SOC, charge transfer and lithium diffusion inside the respective electrodes can also be studied in these time domains.¹⁻⁴ However, since some data points at low (< 1 Hz) or high (> 30 kHz) frequencies exhibited systematic errors in Kramers-Kronig transformation (please also see Answer#32) the evaluated frequency range was confined to 30 kHz – 1 Hz.

The fact that the literature gives such a range, and all others measure it is not a scientific argument. Furthermore, the choice of the range of frequencies is strongly related to the type of chemistry of the battery and, most importantly, the electrochemical processes which occur.

Answer #13:

EIS experiments were carried out to characterize the interfacial transport properties complementary to quantitative insights from NMR spectroscopy. As commonly done in academic research, we consulted previous literature on similar cell chemistries (*i.e.* NMC positive electrodes and lithium metal negative electrodes with liquid electrolytes) to find an appropriate lower frequency limit to obtain insights into the interfacial transport properties. Since 0.1 Hz is often applied as the lower frequency limit, we adapted this in our impedance measurements. While this yielded satisfactory impedance data for interfacial transport and our interpretation of EIS data was in alignment with literature, we were not aware about the validity criteria for EIS data at the time we prepared the initial version of the manuscript. We would like to thank reviewer #3 once again for pointing this out, as this allowed us to exclude compromised data point from our EIS analysis. Nevertheless, we still believe that choosing a

frequency range in agreement with previous literature is a valid approach to establish experimental conditions to begin with. Since the selected frequency range was satisfactory for our interests, no further adjustments were required.

Comment #14:

...For the low frequency impedance, a slight drift is observed over the course of two hours.

I wish the authors had included comments on observed drift and the frequencies that could be represented in the complex plane plots to assist readers who may be less experienced with EIS analysis, particularly given that the reliable measurement range is limited to 30 kHz to 1 Hz.

Answer #14:

The frequency was added to the first data points, which exhibit a deviation of impedance over time. However, due to lithium metal reacting with the electrolyte, absolute stability of impedance data is challenging to accomplish in cell chemistries comprising lithium metal and liquid electrolytes.

Comment #15:

Figures S20, S25, S27: Equivalent circuit fit: The use of the Rs-CPE model in series appears unwarranted. The Kramers-Kronig analysis reveals that the range between 30 kHz and 1 Hz spans only four decades, resulting in a limited number of data points. This scarcity makes it difficult to draw meaningful conclusions or achieve reliable fitting. Furthermore, the absence of data at both lower and higher frequencies complicates the identification and validation of the CPE's presence. Figures S20, S25 and S27 should be removed.

Answer #15:

We agree with Reviewer#3 that the EIS data should be discussed qualitatively rather than quantitatively. Thus, the absolute values for the R-CPE elements (Tables S1-3) have been removed from the supporting information. Despite their uncertainty, we consider the equivalent circuit fits in Nyquist plots a valuable guide for the eye of less EIS experienced readers, which desire to understand the differentiation of electrochemical processes based on the EIS spectra. Thus, we would like to retain figures S20, S25 and S27 in the supporting information. The existing statements on the uncertainty of equivalent circuit fits have been further emphasized in the revised version of the manuscript.

REVIEWERS' COMMENTS

Reviewer #1 (Remarks to the Author):

Comment #1:

Overall, the manuscript has improved since the last iteration. There is still some lack of clarity surrounding the exact nature of the alloy phase formed, due to the reliance on NMR. Likewise, some simple ^{19}F NMR would also show that the SrF₂-PVDF-HFP is not consuming Li.

Answer #1:

We thank the reviewer for acknowledging our additional efforts. We relied on NMR for the characterization since it is highly sensitive and enables a better distinction of signals than XRD, where multiple components can be observed and identifying the alloy signals could be challenging. However, we are unaware how ^{19}F NMR may enable insights into the consumption of lithium species. Even in cases where no novel ^{19}F peak is observed, new Fluorine compounds that consume lithium might form but may not be observed due to the limit of detection.

Comment #2:

I also disagree with the rebuttal statement:

'Echo-based pulse sequences can act as T2 filters for fast relaxing species (e.g. lithium metal), which limits the quantitative interpretation of NMR spectra.'

Spin-echo NMR spectroscopy is routinely used to record wide-line paramagnetic, magnetic, and metallic NMR signals. Counter to the statement above, the lack of echo could mean broader resonances are being filtered during acquisition.

Answer #2:

As displayed in Supplementary Figure 2, lithium metal exhibits comparatively fast transversal relaxation within 1 ms. Therefore, we expected a loss of lithium metal signal intensity throughout the echo pulse sequence (tens of μs), which could limit the quantitative nature of ^7Li NMR experiments. While a spin-echo is indeed helpful for wide-line metallic NMR signals, we prioritized that our protocol remains quantitative, not distorting baseline information.

Reviewer #2 (Remarks to the Author):

Review Report Manuscript 513991_2_art_file_10492359_st6fzz

The authors did further improve their manuscript "Origins of lithium inventory reversibility with alloying functional layers in 'anode-free' lithium metal batteries". Especially the addition of ex-situ and operando NMR experiments to corroborate the formation of an Li_xSn alloying layer on the Cu current collector improved the quality of the manuscript significantly. I recommend publishing after

addressing the following minor remarks:

Comment #3:

1. Previous Fig. S10 was dismissed and as it seems not replaced or replaced by Figure S27? The numbering in the manuscript and in the SI are not consistently changed. Please check carefully.

Answer #3:

Indeed, Figure S10 was only partially replaced by Figure S27 (now Figure S26). With Li alloy signals of low intensity being uncovered by 1st order phase correction of spectra previously depicted in Figure S10, the new graph (Figure S27, respectively S26) focuses on this region instead of the 0 ppm region. In our previous version of the manuscript, the 0 ppm region was discussed in more detail and concluded to not display alloying signals. Since these are now observable by varied processing and additional NMR spectra at higher magnetic field, we considered the discussion of signals in the 0 ppm region as superfluous. The references to all figures in the main manuscript and supporting information were carefully checked and are now accurate.

Comment #4:

2. Page 19 states that MAS NMR of scraped powder from negative electrodes was not feasible. Instead, NMR analyses of complete negative electrodes were conducted. Please clarify that these were done in a wide-line setup (no MAS). This at the moment remains unclear in the manuscript.

Answer #4:

We thank the reviewer for noticing this issue and pointed out that the respective experiments were conducted under static conditions.